# Single cell functional genomics reveals the importance of mitochondria in cell-to-cell phenotypic variation

Riddhiman Dhar[1,2,3], Alsu M Missarova[4], Ben Lehner[1,2,5†]*, Lucas B Carey[4†‡]*

[1]Systems Biology Program, Centre for Genomic Regulation (CRG), The Barcelona Institute of Science and Technology, Barcelona, Spain; [2]Universitat Pompeu Fabra, Barcelona, Spain; [3]Department of Biotechnology, Indian Institute of Technology, Kharagpur, India; [4]Department of Experimental and Health Sciences, Universitat Pompeu Fabra, Barcelona, Spain; [5]Institució Catalana de Recerca i Estudis Avançats (ICREA), Barcelona, Spain

**\*For correspondence:**
lehner.ben@gmail.com (BL);
lucas.carey@pku.edu.cn (LBC)

[†]These authors contributed equally to this work

**Present address:** [‡]Center for Quantitative Biology, Peking-Tsinghua Center for Life Sciences, and the Academy for Advanced Interdisciplinary Studies, Peking University, Beijing, China

**Competing interests:** The authors declare that no competing interests exist.

**Abstract** Mutations frequently have outcomes that differ across individuals, even when these individuals are genetically identical and share a common environment. Moreover, individual microbial and mammalian cells can vary substantially in their proliferation rates, stress tolerance, and drug resistance, with important implications for the treatment of infections and cancer. To investigate the causes of cell-to-cell variation in proliferation, we used a high-throughput automated microscopy assay to quantify the impact of deleting >1500 genes in yeast. Mutations affecting mitochondria were particularly variable in their outcome. In both mutant and wild-type cells mitochondrial membrane potential – but not amount – varied substantially across individual cells and predicted cell-to-cell variation in proliferation, mutation outcome, stress tolerance, and resistance to a clinically used anti-fungal drug. These results suggest an important role for cell-to-cell variation in the state of an organelle in single cell phenotypic variation.

DOI: https://doi.org/10.7554/eLife.38904.001

## Introduction

Isogenic populations often exhibit considerable phenotypic heterogeneity even in an identical environment. One common phenotypic variation that has been observed in isogenic populations of microbial and mammalian cells, including cancer cells is variation in proliferation rate (*Sandler et al., 2015*; *Fridman et al., 2014*; *Levy et al., 2012*; *Yaakov et al., 2017*; *Ferrezuelo et al., 2012*; *Polymenis and Schmidt, 1999*; *Gupta et al., 2011*; *Kiviet et al., 2014*; *Roesch et al., 2010*). Phenotypic variations that are often coupled with variation in proliferation rate are the abilities of an individual cell to survive stress and drug treatment (*Levy et al., 2012*; *Yaakov et al., 2017*). In this regard, the existence of 'persister' cells in microbial populations is well known and poses a significant challenge for antibiotic treatment (*Fridman et al., 2014*; *Balaban et al., 2004*; *Kussell et al., 2005*; *Wakamoto et al., 2013*; *LaFleur et al., 2006*; *Bojsen et al., 2017*). Similarly, individual cells in tumors have been shown to vary in their ability to survive anticancer drugs and can lead to drug-resistant populations (*Shaffer et al., 2017*; *Sharma et al., 2010*; *Ramirez et al., 2016*; *Hata et al., 2016*; *Rego et al., 2017*; *Márquez-Jurado et al., 2018*). Recent advances in single-cell techniques are revealing the extent of transcriptomic and metabolic differences among isogenic cells (*Dey et al., 2015*; *Trapnell, 2015*). The existence of such heterogeneity in gene expression in isogenic microbial and animal populations has been shown – to some extent – to underlie the variable outcome of mutations (*Dickinson et al., 2016*; *Burga et al., 2011*; *Raj et al., 2010*; *Eldar et al., 2009*; *Horvitz and Sulston, 1980*). Incomplete mutation penetrance and variable expressivity is also

common in human disease (*Cooper et al., 2013*; *Zlotogora, 2003*; *Giudicessi and Ackerman, 2013*; *Otterson et al., 1999*).

Heterogeneity can arise due to stochastic fluctuations in biological processes taking place inside cells. This can happen due to the small numbers of molecules involved in processes such as transcription (*Berg, 1978*; *Rigney, 1979*; *Cookson et al., 2010*) or during stochastic partitioning of cellular components during cell division (*Birky and Skavaril, 1984*; *Huh and Paulsson, 2011*). Although genetic variation has been shown to influence proliferation heterogeneity (*Ziv et al., 2013*) and cell-to-cell variation in the expression level of single genes and levels of metabolite has been correlated with variation in proliferation rate and stress and drug resistance (*Levy et al., 2012*; *Yaakov et al., 2017*; *Shaffer et al., 2017*; *Burga et al., 2011*; *Raj et al., 2010*; *Eldar et al., 2009*; *Li et al., 2018*; *Rotem et al., 2010*; *Battich et al., 2015*), the true underlying causes of such phenotypic heterogeneity are poorly understood.

To identify genes and cellular processes involved in the generation of phenotypic heterogeneity we set up a high-throughput microscopy assay to quantify proliferation heterogeneity in a yeast population. Using this assay, we quantify the impact of deletion of >1500 genes on proliferation heterogeneity. We present evidence that the variation in mitochondrial membrane potential is an important determinant of phenotypic heterogeneity in individual cells. We also show that mitochondrial membrane potential impacts gene expression and stress tolerance and drug resistance in individual cells. Taken together, our work suggests an important role for an organelle in generating phenotypic heterogeneity across individual cells in a homogenous environment.

## Results

### Natural and lab yeast populations show proliferation heterogeneity

To investigate cell-to-cell variation in proliferation rates, we set up a high-throughput automated time-lapse microscopy assay that measures the proliferation rates of thousands of single-cells per plate as they grow into micro-colonies. The assay uses a microscope with laser-based autofocus for image acquisition and a liquid handling robot to minimize density-dependent effects on proliferation. The data obtained are highly reproducible with mode proliferation rate of a lab strain being $0.407 \pm 0.011 \text{ h}^{-1}$, (mean ±sd) during >2 years of data collection (n = 44 batches; *Figure 1A*).

Laboratory strains of the budding yeast *Saccharomyces cerevisiae* showed substantial cell-to-cell variation in proliferation, with ~10% of cells forming a slow growing sub-population in defined growth medium (*Figure 1A*) (*Levy et al., 2012*; *Ziv et al., 2013*). This slow growing sub-fraction is not unique to laboratory strains but exists in all natural and clinical isolates that we tested (*Figure 1B*; *Supplementary file 1*) (*Ziv et al., 2013*). Growth of the culture for an additional 20 generations did not alter the proliferation rate distribution; the mixture of slow and fast proliferating cells is maintained (*Figure 1C*). Proliferation is therefore a stable heterogeneous phenotype within a population, with the amount of heterogeneity depending on the genetic background.

### A genome-scale screen to identify genes that alter proliferation heterogeneity

The effect of individual gene deletions on population-level growth rate has been well studied (*Giaever et al., 2002*; *Baryshnikova et al., 2010*). Many deletions have been shown to reduce population growth rate and can do so in different ways. Deletions can uniformly affect fitness of all the cells or alternatively, can affect fitness of a sub-population whereas the rest of the population remains unaffected. Inter-individual variation in the outcome of mutations has been observed before in multicellular organisms (*Dickinson et al., 2016*; *Burga et al., 2011*; *Raj et al., 2010*) but its relative occurrence has not been systematically quantified.

We therefore used the automated microscopy assay to quantify proliferation rate heterogeneity in triplicate for 1600 gene deletion mutants (*Supplementary file 2*, including 1150 gene deletions previously reported as affecting growth rates [*Giaever et al., 2002*; *Baryshnikova et al., 2010*]). We obtained reproducible data (where at least two replicate measurements showed good agreement) for 1520 deletions, with 1112 of these reducing the population proliferation rate in our experiment (Mann-Whitney U test, FDR < 0.1).

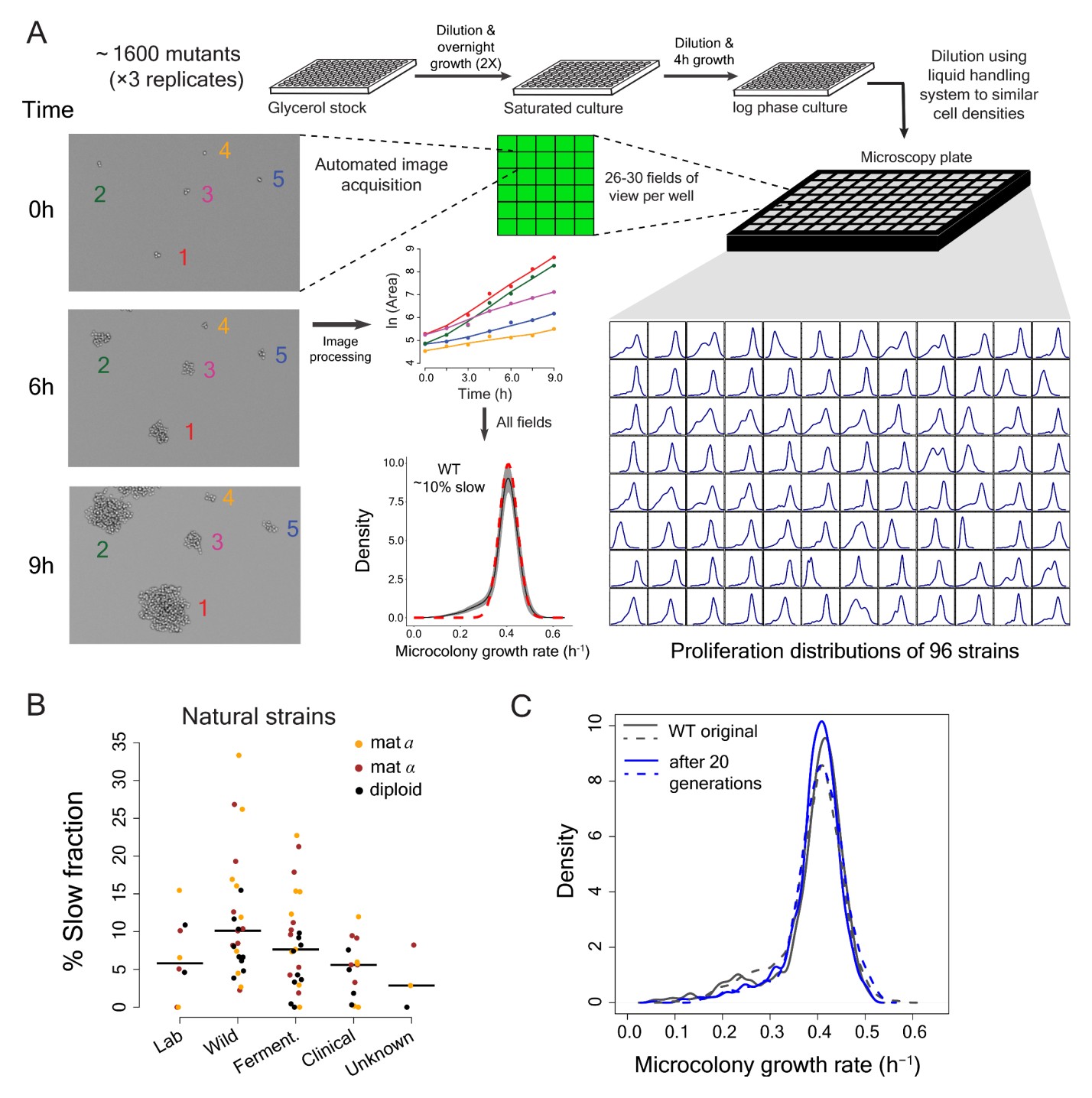

**Figure 1.** High-throughput analysis of single cell proliferation rate heterogeneity. (**A**) High throughput microscopy setup – log phase yeast cells were diluted onto conA coated microscopy plate using Biomek NX liquid handling system to have similar cell density across wells. Cells were observed using an ImageXpress Micro system. Images were processed using custom scripts and data for area of microcolony vs. time were obtained. The points in the area vs. time graph show actual data and the solid lines show lowess fits. Data collected from all fields of view in a well constitute a microcolony proliferation rate distribution for a strain. The common lab yeast strain BY4741 (WT) has ~10% slow proliferating sub-population. The density shows mean density and the shaded areas in grey represent ±1 s.d. value at each point. The dotted red line shows the expected proliferation distribution if it were normally distributed. (**B**) Natural strains of yeast (*Ziv et al., 2013*) also have slow proliferating sub-populations. Each point represents data for one strain. Solid lines show median value. (**C**) WT strain re-created the original proliferation distribution even after 20 generations of growth. The plot shows data from two replicate measurements.

*Figure 1 continued on next page*

*Figure 1 continued*

DOI: https://doi.org/10.7554/eLife.38904.002

The following source data is available for figure 1:

**Source data 1.** Percentage of slow-growing cells in natural yeast strains.

DOI: https://doi.org/10.7554/eLife.38904.003

Deletion strains with similar population proliferation rates often showed strikingly different degrees of intra-population heterogeneity (Fig, 2A-C). At the single cell level, ~39% of all mutants with a significant reduction in population proliferation rate (1112 mutants) showed significantly higher variation in mutation outcome compared to the WT strain (Mann-Whitney U test, FDR < 0.1, after correcting for change in mode growth rate). Among these mutants, ~13% had the same mode growth rate as the WT strain while showing higher variability (Mann-Whitney U test, FDR < 0.1). However, almost all mutants (1111 of 1112 mutants) had a subset of cells proliferating at the same rate as the bulk of the wild-type (WT) population (one sample Wilcoxon rank-sum test for overlap with bulk WT distribution differing from zero, FDR < 0.1; *Figure 2D*, *Figure 2—figure supplement 1*). Thus, a highly variable outcome is actually the normal outcome for proliferation rate at the single cell level when a non-essential gene is inactivated (*Figure 2D*, *Figure 2—figure supplement 2A*).

## Deletion of genes involved in mitochondrial function alter heterogeneity

To identify the determinants of this cell-to-cell variation in growth-rate and mutational impact we classified each of the deletions by how it affected both the mode and distribution of cellular proliferation rates (*Supplementary file 2*, *Figure 2A,B*). Approximately, 17% of the mutants showed no change in either mode proliferation rate or percentage of slow sub-population (in grey), whereas ~43% exhibited a change in mode proliferation rate but no change in slow fraction (in light blue). Interestingly, 48 mutants reduced the slow fraction without any change in mode proliferation rate (in red) and 97 mutants increased the slow fraction without altering the mode proliferation rate (in blue). In addition, there were 78 mutants that reduced both the slow fraction and the mode proliferation rate (in orange). Finally, 370 mutants reduced the mode growth rate but increased the slow sub-population (in magenta, *Figure 2A*). Across mutants, we observed a strong inverse relationship between mean growth rate and noise (co-efficient of variation) (*Figure 2C*), as has been observed for gene expression (*Newman et al., 2006*; *Bar-Even et al., 2006*).

To identify biological processes associated with changes in the slow growing sub-population, we performed a GO functional enrichment analysis on genes in these categories (FDR < 0.1). Deletions causing the largest increase in the fraction of slow proliferating cells were highly enriched for nuclear genes encoding mitochondrial proteins (*Figure 2C,E*). Among the mutants that increased the slow fraction but also reduced mode growth rate (*Figure 2E*, magenta), ~30% localized to mitochondria (~1.2 fold enrichment), ~13% localized to the mitochondrial envelope (>1.6 fold enrichment) and ~4.6% were involved in cellular respiration (~2-fold enrichment). In particular, deletion of genes that localized to the mitochondrial envelope resulted in a large increase in slow fraction and noise (*Figure 2E*, *Figure 2—figure supplement 2B*, *Supplementary file 2*). Mutations that affect mitochondria, and in particular the mitochondria membrane, increase heterogeneity, suggesting that heterogeneity in proliferation might be associated with cell-to-cell variation in mitochondria. Among genes with other functional associations, deletion of genes with kinase activity (STE11, SNF1, NPR1, ATG1, HXK2), genes with cytoskeletal protein binding capability (HOF1, SRV2, SHE1) and genes associated with carbohydrate transport (SNF3, HXK2) reduced percentage of slow growing cells but did not alter the mode growth rate.

## Mitochondrial membrane potential but not amount predicts slow growth

To further investigate the role of mitochondria in proliferation heterogeneity, we used the Mito-Tracker dye to quantify mitochondrial amount in WT cells and five deletion strains with very different proliferation distributions. Total mitochondria amount varied little across the strains (*Figure 2—figure supplement 2C*), ruling out cell-to-cell variation in segregation of the organelle as a driver of

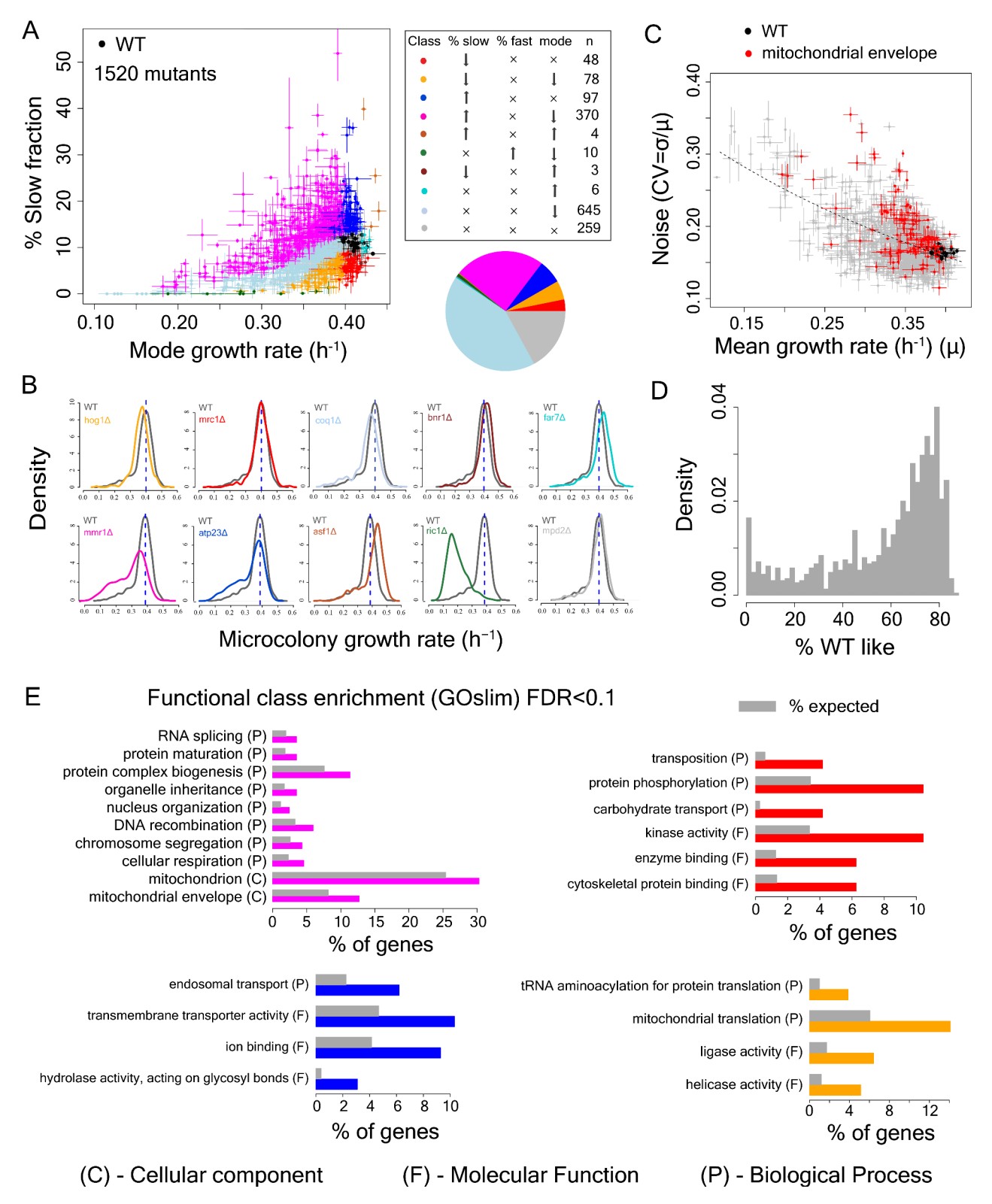

**Figure 2.** Single cell proliferation rate distributions for 1500 gene deletions. (**A**) Mode growth rate (h$^{-1}$) and % slow fraction for 1520 deletion strains. The points represent average values across replicates and the bars represent ±1 s.d. values. The colours show classification of mutants into different categories according to change in mode growth rate (see Materials and methods, FDR < 0.1) and change in % slow fraction (FDR < 0.1) compared to the wild-type (WT) strain. The table and pie chart show the number and proportion of strains in each group (colour coded). Replicate data for WT strain

*Figure 2 continued on next page*

*Figure 2 continued*

are shown by multiple black points. (B) Examples of growth distributions of mutants classified into different groups which are colour coded as in A. The distribution in dark grey shows WT growth distribution. (C) Coefficient of variation (CV) vs. mean growth rate for all strains. WT values are shown in black; mutants of genes that localize to mitochondrial envelope in red. The points represent average values across replicates and the bars represent ±1 s.d. values. (D) % of WT-like cells in all mutants showing variable mutation outcome. It was calculated for all mutants showing significant reduction in mean proliferation rate and had significant proportion of cells growing as fast as the bulk of the WT proliferation distribution (Wilcoxon rank-sum test). (E) Functional class enrichment (GOslim) analysis for different classification groups show significantly enriched functional classes (hypergeometric test, FDR < 0.1). P – Biological Process, F – Molecular Function, C- Cellular Component. Bars show % of genes in a particular group (colour coded) being present in that particular functional class.

DOI: https://doi.org/10.7554/eLife.38904.004

The following source data and figure supplements are available for figure 2:

**Source data 1.** Percentage of slow-growing cells in WT and mutant strains.
DOI: https://doi.org/10.7554/eLife.38904.007
**Figure supplement 1.** Schematic diagram showing calculation of %WT like cells from mutant proliferation distributions.
DOI: https://doi.org/10.7554/eLife.38904.005
**Figure supplement 2.** Calculation of slow-growing sub-population and functional enrichment analysis.
DOI: https://doi.org/10.7554/eLife.38904.006

heterogeneity. However, signal from the mitochondria membrane potential dye TMRE varied substantially across WT, mutants and natural strains (*Figure 3A,B*). This suggested that the mitochondria membrane potential – but not their amount – might be driving proliferation heterogeneity.

To determine if mitochondrial membrane potential is correlated with variation in growth rate within a population, we sorted wild-type cells according to their TMRE staining and measured the fraction of slow proliferating cells. The population with high TMRE was highly enriched for slowly proliferating cells (*Figure 3C*). This same fraction was also strongly enriched for respiration deficient cells (*Figure 3D*). This strong enrichment for slow proliferation and respiration deficiency was also observed for the cells with high TMRE in gene deletion strains and in natural isolates (*Figure 3—figure supplement 1A–D*).

We further quantified the TMRE signal and proliferation distribution in a set of twelve strains that differ only by naturally occurring polymorphisms known to affect mitochondrial function by altering mtDNA inheritance (*Dimitrov et al., 2009*). Across all datasets, the percentage of slowly proliferating cells showed a high correlation with the percentage of respiration-deficient cells (*Figure 3E,F*) as well as with the percentage of high TMRE cells (*Figure 3—figure supplement 2A*).

Although cell-to-cell variation in mitochondrial amount did not predict proliferation rate variation (*Figure 3—figure supplement 2B*), mtDNA copy number was substantially lower in the cells with high TMRE (*Figure 4A*; *Figure 4—figure supplement 1A,B*), suggesting a likely role of mtDNA copy number in defining mitochondrial membrane potential and ultimately, in generation of growth rate heterogeneity.

To establish a causal relationship between mtDNA copy number and slow growth, we introduced an extra copy of the mitochondrial DNA polymerase Mip1 (*Genga et al., 1986*), which increased mtDNA copy number 3-fold (*Figure 4B*). This reduced both the fraction of slow proliferating and respiration-deficient cells and the fraction of cells with high TMRE signal (*Figure 4C,D*), suggesting that variation in mtDNA copy number can be causal for variation in both mitochondrial membrane potential and proliferation. Consistent with an effect of mtDNA copy number on growth, knocking out of Mip1 gene led to complete loss of mtDNA and resulted in completely slow growing yeast population compared to WT (*Figure 4—figure supplement 1C,D*) (*Genga et al., 1986*). Furthermore, mtDNA copy number showed a strong negative correlation with the percentage of slow proliferating cells across mutants (*Figure 4E*, *Figure 4—figure supplement 1E*). Finally, forcing cells to respire by pre-growing them on ethanol as the sole carbon source prior to growth in glucose decreased the fraction of slowly proliferating cells (*Figure 4F*). Taken together, these results suggest that alterations in mitochondrial membrane potential, which can be caused by mtDNA copy number reduction below a threshold and other mechanisms is the underlying cause of slow growth in individual cells.

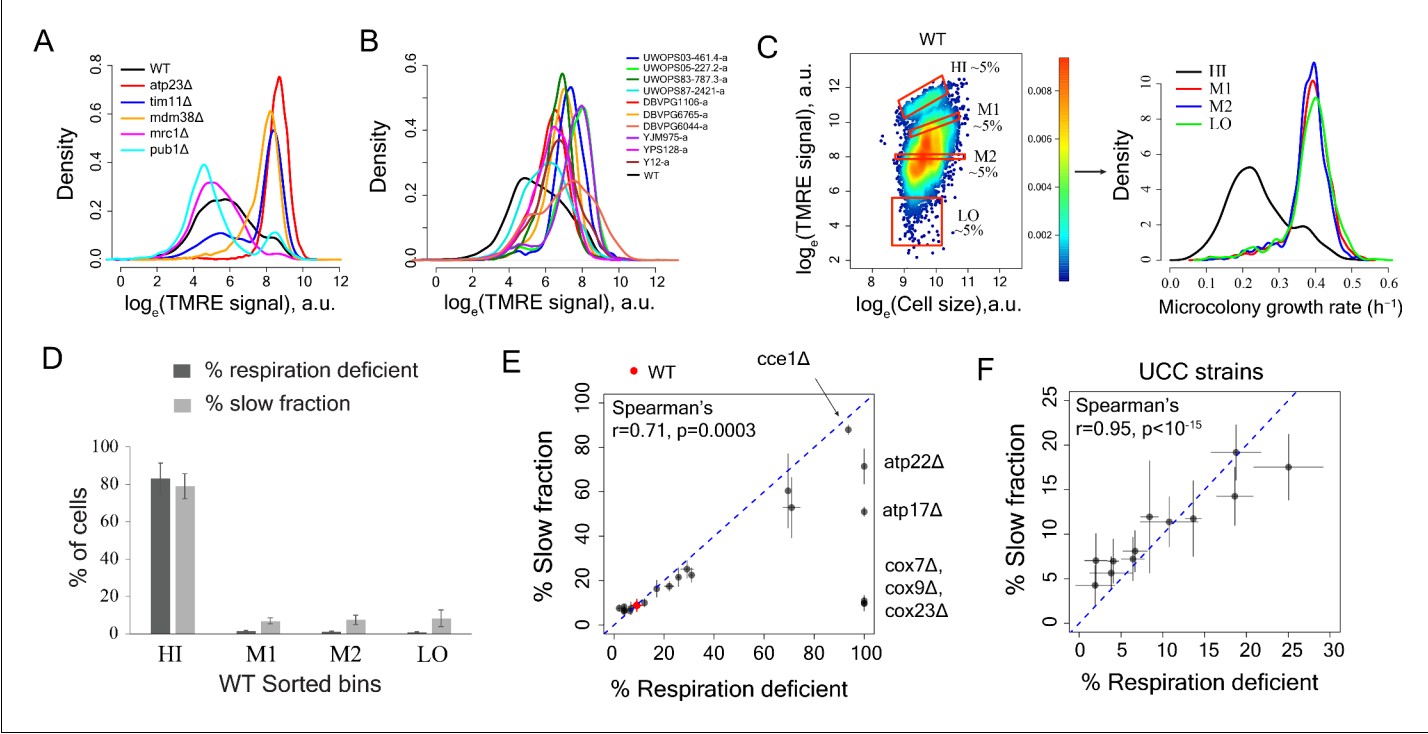

**Figure 3.** Variation in mitochondria potential across single cells underlies proliferation heterogeneity. (**A**) TMRE stain intensity (log transformed) measured by flow cytometry in WT and deletion mutants. (**B**) TMRE intensity in WT and natural isolates of *S. cerevisiae* strains. (**C**) WT cells were sorted by TMRE signal intensity into four bins HI, M1, M2 and LO with gates as shown (~5% of the population sorted in each bin) and growth rate distributions were measured using high throughput microscopy setup. HI bin was enriched for slow growing cells. (**D**) % of respiration deficient cells in each bin from WT strain. The columns represent the average values from 12 independent experiments and the bars show ±1 s.d. values. (**E**) Percentage of respiration deficient cells in WT and mutant strains is positively correlated with the percentage of slow growing cells. The blue dotted line represents y = x line. The error bars represent ±1 s.d. measured from at least two biological replicates for each strain. (**F**) Percentage of respiration deficient cells in UCC strains (*Dimitrov et al., 2009*) is strongly positively correlated with the percentage of slow growing cells. The blue dotted line represents y = x line. The error bars represent ±1 s.d. measured from at least two biological replicates for each strain.

DOI: https://doi.org/10.7554/eLife.38904.008

The following source data and figure supplements are available for figure 3:

**Source data 1.** TMRE intensity distribution in WT and mutant strains.
DOI: https://doi.org/10.7554/eLife.38904.011

**Figure supplement 1.** Cells with high mitochondrial membrane potential in mutant yeast strains, natural yeast isolates and diploid strain BY4743 show enrichment for slow-growing and respiration deficient cells.
DOI: https://doi.org/10.7554/eLife.38904.009

**Figure supplement 2.** High TMRE signal but not mitotracker green signal predicts slow-growing subpopulation
DOI: https://doi.org/10.7554/eLife.38904.010

## Variation in mitochondrial membrane potential predicts additional phenotypic heterogeneity including drug resistance

To systematically understand the physiological differences between sub-populations that vary by mitochondrial membrane potential, we analyzed the transcriptome by RNA sequencing. Cells with high TMRE have low expression of respiratory and proliferation-associated genes (*Figure 5A*, *Figure 5—figure supplement 1A,B*). Consistent with previous analyses of slow proliferating cells (*Yaakov et al., 2017*; *van Dijk et al., 2015*), they also exhibit a DNA damage response (*Figure 5B*) and signs of iron starvation (*Figure 5C*), which has previously been reported for respiration deficient cells (*Veatch et al., 2009*; *Puig et al., 2005*). Cells with intermediate TMRE have very similar proliferation distributions to cells with low potential (*Figure 3C*). However, their gene expression was

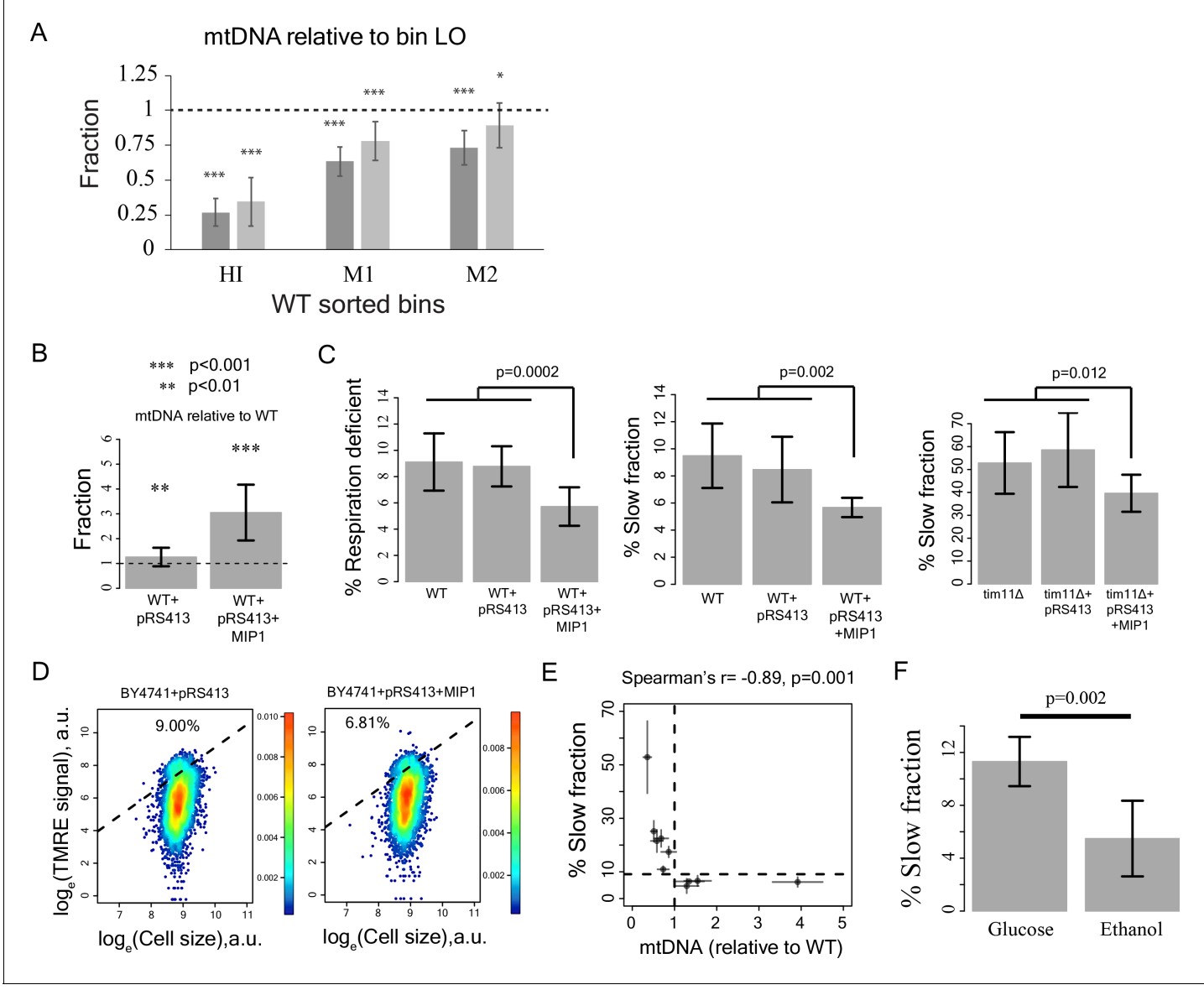

**Figure 4.** Reduction in mtDNA copy number causes slow growth. (**A**) mitochondrial DNA (mtDNA) copy number in the sorted bins HI-LO from WT strain measured through quantitative PCR. Two columns show results from two independent experiments. The column represents average mtDNA copy number calculated based on five pairs of primers binding mtDNA and five pairs of primers binding nuDNA and three technical replicates for each of these primers. The bars show ±1 s.d. values. (**B**) Overexpression of Mip1 gene in WT strain led to significant increase in mtDNA copy number (**C**) Overexpression of Mip1 gene led to significant reduction in percentage of respiration deficient cells and in slow growing subpopulation in WT strain and tim11Δ mutant. Data are from at least four biological replicates. (**D**) Overexpression of MIP1 gene in WT strain reduced percentage of cells with high TMRE signal. (**E**) Percentage of slow growing sub-population was strongly correlated with mtDNA copy number in mutant strains. The dotted lines represent values for WT strain. The error bars represent ±1 s.d. values. (**F**) Pre-growing WT strain overnight in medium containing ethanol as sole carbon source (that required respiration) reduced percentage of slow growing sub-population by ~50% compared to pre-growth in medium containing glucose as the sole carbon source. Data are from six biological replicates.
DOI: https://doi.org/10.7554/eLife.38904.012

The following source data and figure supplement are available for figure 4:

**Source data 1.** mtDNA copy number, % slow fraction and % respiration deficient cells in WT and mutant strains.
DOI: https://doi.org/10.7554/eLife.38904.014
**Figure supplement 1.** Role of mtDNA in generating slow growth and high mitochondrial membrane potential .
DOI: https://doi.org/10.7554/eLife.38904.013

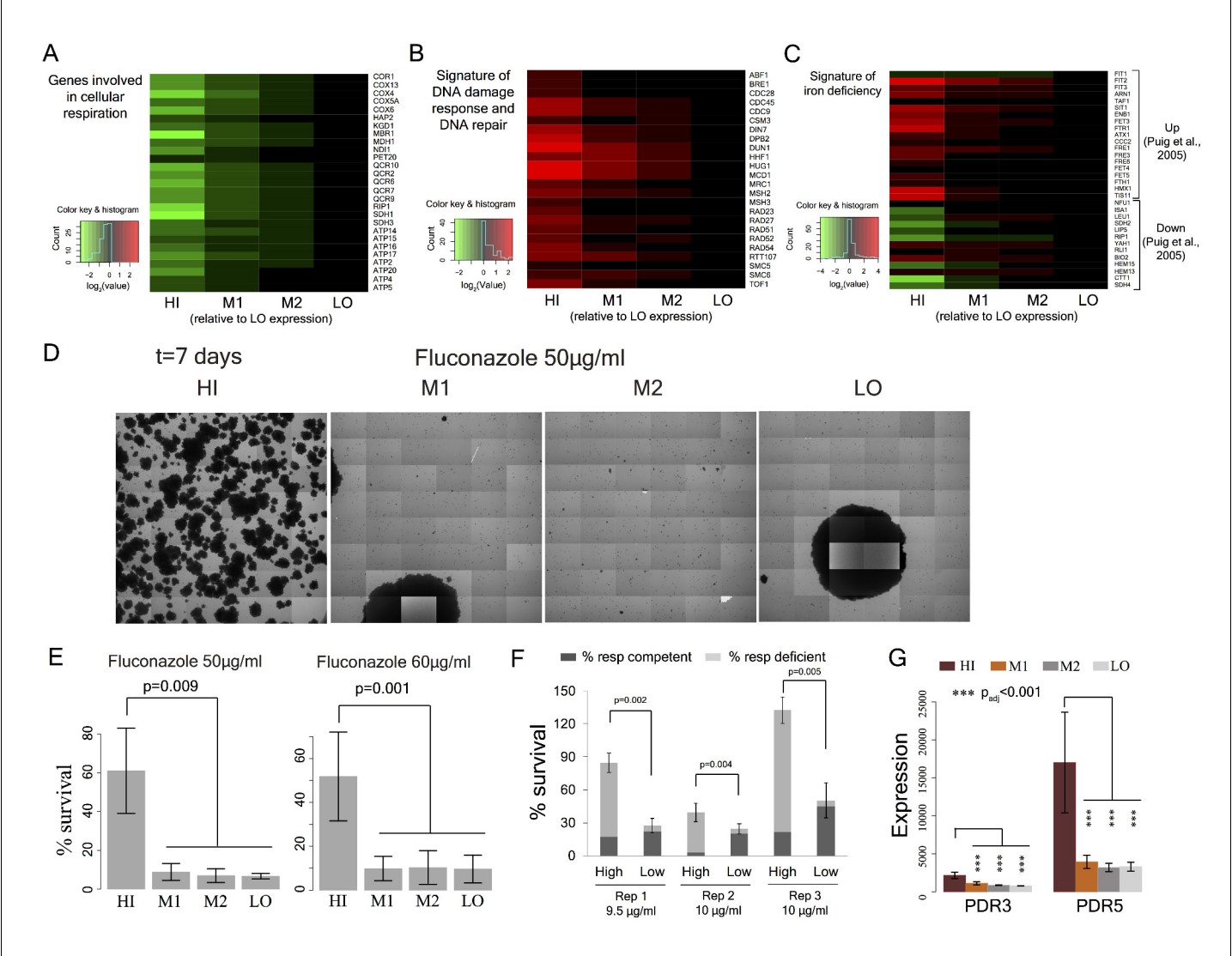

**Figure 5.** Cell-to-cell variation in mitochondria potential predicts single cell drug resistance. (A) Heatmap of expression of respiration genes in cells sorted by their TMRE signal intensity (bins HI-LO). (B) Heatmap of expression of DNA damage response and DNA repair genes. (C) Heatmap of expression of genes associated with iron deficiency (**Puig et al., 2005**; **Veatch et al., 2009**). Data are from four independent experiments. (D) Sorted bins from WT cells were grown in a commonly used antifungal drug fluconazole and were observed under microscope for growth over 7 days. The images show growth of cells in bins HI, M1, M2 and LO in 50 μg/ml of fluconazole after 7 days. (E) Cells of HI bin showed significantly higher survival compared to other bins in both 50 μg/ml (three independent experiments) and 60 μg/ml fluconazole (four independent experiments). Cells were grown in liquid medium supplemented with fluconazole on microscopy plates and viability was calculated from microscopic observations over 7 days. Colonies showing growth rate above 0.02 h$^{-1}$ after first time point were considered to be survivors. Error bars show ±1 s.d. values. (F) Percentage survival of high and low TMRE cells on fluconazole plates. High TMRE cells showed higher survival than low TMRE cells (Mann-Whitney U test). A substantial fraction of surviving high TMRE cells were respiration competent. The error bars represent ±1 s.d. values from six technical replicates for each bin. X-axis shows fluconazole concentrations used from three independent experiments. (G) From RNA sequencing data, cells from HI bin showed significantly higher expression of multidrug transporter *PDR5* gene and its transcriptional activator *PDR3* compared to cells from bins M1, M2 and LO. Results are from four independent experiments.

DOI: https://doi.org/10.7554/eLife.38904.015

The following source data and figure supplements are available for figure 5:

**Source data 1.** Transcriptomic changes and increased antifungal resistance in high TMRE cells.
DOI: https://doi.org/10.7554/eLife.38904.028

**Figure supplement 1.** Expression patterns of diverse gene functional classes in cells with low, medium and high mitochondrial membrane potential.
DOI: https://doi.org/10.7554/eLife.38904.016

**Figure supplement 2.** Stress survival and expression of stress response genes in cells with different levels of mitochondrial membrane potential.

*Figure 5 continued on next page*

*Figure 5 continued*

DOI: https://doi.org/10.7554/eLife.38904.017
**Figure supplement 3.** Molecular pathways that underlie increased drug resistance in cells with high mitochondrial membrane potential.
DOI: https://doi.org/10.7554/eLife.38904.018
**Figure supplement 4.** Growth lag and slowdown in high-throughput microscopy assay.
DOI: https://doi.org/10.7554/eLife.38904.019
**Figure supplement 5.** Correlation of growth rates of deletion mutants within experimental replicates and with published datasets.
DOI: https://doi.org/10.7554/eLife.38904.020
**Figure supplement 6.** Test for recovery of growth rate, mtDNA copy number and respiration capability in slow-growing sub-population.
DOI: https://doi.org/10.7554/eLife.38904.021
**Figure supplement 7.** Slow to fast switching in sorted sub-populations and in gene deletion mutants.
DOI: https://doi.org/10.7554/eLife.38904.022
**Figure supplement 8.** Percentage of microcolonies showing slow to fast switching and fast to slow switching in the sub-population of cells in TMRE sorted bins HI, M1, M2 and LO from WT strain using different growth rate cutoffs (0.24 $h^{-1}$, 0.26 $h^{-1}$, 0.28$h^{-1}$, 0.30$h^{-1}$, 0.32$h^{-1}$, 0.34$h^{-1}$, and 0.36 $h^{-1}$).
DOI: https://doi.org/10.7554/eLife.38904.023
**Figure supplement 9.** Percentage of microcolonies showing slow to fast switching and fast to slow switching in the sub-population of cells in TMRE sorted bins HI, M1, M2 and LO from WT strain using alternative growth rate cutoffs as shown in the figure.
DOI: https://doi.org/10.7554/eLife.38904.024
**Figure supplement 10.** Measurement of TMRE in cells of sorted bins HI-LO from WT strain.
DOI: https://doi.org/10.7554/eLife.38904.025
**Figure supplement 11.** Microcolony growth rate distribution for cells in HI bin from TMRE sorted WT strain for calculation of switching rate from high to low TMRE state.
DOI: https://doi.org/10.7554/eLife.38904.026
**Figure supplement 12.** Reproducibility of RNAseq experiments and growth rate measurements.
DOI: https://doi.org/10.7554/eLife.38904.027

substantially different (*Figure 5—figure supplement 1C,D*), including reduced expression of respiratory genes in cells with intermediate TMRE (*Figure 5—figure supplement 1D*).

In bacteria (*Ihssen and Egli, 2004*) and yeast (*Levy et al., 2012*; *Brauer et al., 2008*; *Lu et al., 2009*), slow growing cells can have increased stress resistance. We therefore tested whether the cells with high TMRE in a population are more resistant to acute heat stress. However, cells with high TMRE were more sensitive to heat shock as well as to oxidative stress (*Figure 5—figure supplement 2A,B*) and expressed some stress-response genes at lower levels (*Figure 5—figure supplement 2C*).

Slow growing microbes and cancer cells often have increased drug resistance (*Fridman et al., 2014*; *Yaakov et al., 2017*; *Wakamoto et al., 2013*; *Ramirez et al., 2016*; *Levin-Reisman et al., 2017*; *Moore et al., 2012*). Moreover, in several species of fungi, complete loss of mitochondria is associated with elevated tolerance to some antifungal drugs (*Hallstrom and Moye-Rowley, 2000*; *Brun et al., 2004*; *Zhang and Moye-Rowley, 2001*) and respiration deficient strains are often isolated from drug-treated patients (*Bouchara et al., 2000*; *Ferrari et al., 2011*; *Peng et al., 2012*).

We tested therefore whether the high TMRE cells differed in their sensitivity to a clinically used antifungal drug, fluconazole. We found that cells with high TMRE were ~5–7 fold more resistant to high concentrations of fluconazole (*Figure 5D,E*; *Figure 5—figure supplement 3A*). The cells surviving fluconazole treatment included a sub-fraction able to respire (*Figure 5F*). Cell-to-cell variation in mitochondrial membrane potential is therefore also an important predictor of cell-to-cell variation in drug resistance.

Fluconazole targets the cytochrome P450 14α-sterol demethylase enzyme (ERG11), resulting in depletion of ergosterol, a key component of the yeast cell membrane (*Hitchcock et al., 1990*; *Ghannoum and Rice, 1999*). Survival in fluconazole has been previously reported to depend on the multidrug transporter PDR5 (*Brun et al., 2004*; *Parsons et al., 2004*; *Miranda et al., 2010*). High TMRE cells had significantly higher level of PDR5 expression (*Figure 5G*; *Figure 5—figure supplement 3B*) as well as higher expression of the ergosterol biosynthesis pathway (*Figure 5—figure supplement 3D*). Consistent with previous work (*Katzmann et al., 1994*; *Delaveau et al., 1994*), the elevated expression of PDR5 in the high membrane potential cells was dependent on the PDR3 transcription factor (*Figure 5—figure supplement 3C*). Thus, the increased resistance in fluconazole of

high TMRE cells is likely to be mediated, at least in part, by increased expression of a multidrug transporter.

## Discussion

In summary, we have shown here that mitochondrial membrane potential – but not the amount of mitochondria – varies substantially across individual yeast cells and that this is associated with cell-to-cell heterogeneity in proliferation, mutation outcome, and stress and drug resistance. Laboratory strains of yeast have long been known to generate respiratory deficient 'petite' colonies at quite high frequency (*Dimitrov et al., 2009*; *Nagley and Linnane, 1970*; *Evans et al., 1985*). However, a slow growing sub-population of cells was observed in all the laboratory, natural, and clinical strains that we tested (*Figure 1A,B*).

Although, mitochondrial genes showed the strongest enrichment for an increased slow fraction in our gene deletion screen, other causes of slow growth will, of course, also exist. For example, deletions of genes associated with chromosome segregation and nucleus organization also affected heterogeneity but had no apparent relation to mitochondrial function. Environmental conditions, both during growth prior to microscopy, and during microscopy, are likely to affect heterogeneity; pregrowth in ethanol reduces proliferation heterogeneity (*Figure 4F*), presumably due to selection for cells with well-functioning mitochondria and high mtDNA copy number.

Previous theoretical studies have proposed that variability in the partitioning of cellular components could lead to heterogeneity (*Birky and Skavaril, 1984*; *Huh and Paulsson, 2011*). However, prior experimental work on the fidelity of mitochondria inheritance has shown it to be high, suggesting it is likely to be of little phenotypic consequence for single cells (*Jajoo et al., 2016*). In contrast, we have shown here that cell-to-cell variation in the *state* of the organelle can be high and predicts phenotypic variation among single cells. Although variation in mitochondrial membrane potential has been shown to exist in isogenic yeast populations (*Fehrmann et al., 2013*), here we have linked such variations to heterogeneity in growth and drug resistance phenotypes. Variation in mitochondrial membrane potential was related to variation in mtDNA copy number in individual cells, but we do not currently know if this is the only – or even the most common – cause of variation in the state of the organelle across single cells. Future work will be required to track down the upstream, proximal causes of this cell-to-cell variation in organelle functional state. The list of gene deletions that alter growth heterogeneity that we have reported here provide a rich resource for this future work.

In addition to the TMRE method we used, three different methods have now been used to identify slow-proliferating yeast within a population: GFP tagged TSL1, GFP tagged HSP12, and FitFlow, a direct method (*Levy et al., 2012*; *Li et al., 2018*; *van Dijk et al., 2015*). These methods agree on some aspects and disagree on others. For example, early work on proliferation heterogeneity in yeast has correlated expression noise in a stress response gene with proliferation heterogeneity (*Levy et al., 2012*) with this ultimately linked to the variation in the activity of the Ras/cAMP/protein kinase A (PKA) pathway and its target transcription factors MSN2 and MSN4 in individual yeast cells leading to variation in expression of downstream stress response genes (*Li et al., 2018*). Some of our results are identical to that of Li *et al.* (*Li et al., 2018*); we also find that *pde2* and *ira2* cells have a reduced fraction of slow growing microcolonies. Others differ. Importantly, TSL1 mRNA expression is lower in the high TMRE sub-population, as are the MSN2/MSN4 targets CTT1, DDR2 and HSP12.

Other studies have suggested a role for the DNA damage response in the generation and or maintenance of proliferation heterogeneity (*Yaakov et al., 2017*; *van Dijk et al., 2015*). Results from our screen show that the slow growing sub-populations in an isogenic yeast population are primarily driven by loss of mitochondrial function in lab, natural and clinically-isolated yeast strains. Loss of mitochondrial function leads to a loss of respiration capability and induced DNA damage response, consistent with previous reports (*Yaakov et al., 2017*; *van Dijk et al., 2015*). However, while it is tempting to speculate that the mitochondria defects in the high TMRE sub-population cause the DNA damage seen in other studies, the partial conflict in the mRNA-sequencing data suggest that this is not the case.

Taken together, results across all four studies suggest that there are multiple types of slow-proliferating cells. It will be interesting to determine the full extent of the heterogeneity in cell-state and stress resistance within isogenic populations.

Prior work on the causes of variation in proliferation rates, stress and drug resistance, and mutation outcome across individuals and individual cells has focused on fluctuations in gene expression as causative influences (*Levy et al., 2012*; *Yaakov et al., 2017*; *Shaffer et al., 2017*; *Burga et al., 2011*; *Raj et al., 2010*; *Eldar et al., 2009*; *Li et al., 2018*; *Rotem et al., 2010*; *Battich et al., 2015*). Here we have shown that, in yeast, variation in an organelle is strongly associated with heterogeneity in gene expression across single cells. In animals, inherited and somatic genetic variation in the mitochondrial genome can act as an important modifier of phenotypic variation (*Haag-Liautard et al., 2008*; *Kujoth et al., 2007*; *Tyynismaa and Suomalainen, 2009*; *Kauppila et al., 2016*). Recent work has also revealed substantial variation in mtDNA copy number across human tumors (*Reznik et al., 2016*). Moreover, in mammalian cells, mitochondrial variability has been suggested to be an important influence on cell-to-cell variation in gene expression and splicing (*Johnston et al., 2012*; *Guantes et al., 2015*; *Guantes et al., 2016*) and to influence variability in cell death by modulating apoptotic gene expression (*Márquez-Jurado et al., 2018*). Taken together, these results suggest important roles for cellular organelles, in general, and mitochondria, in particular, in the generation of heterogeneity among individual cells. In future work, therefore, it will be important to test the extent to which cell-to-cell variation in the state of mitochondria and other organelles also contributes to variable phenotypic outcomes, mutation effects, and drug resistance in human cells, including in cancer.

## Materials and methods

### High-throughput microscopy assay

Our high-throughput microscopy assay was inspired by the microcolony growth measurement assay by *Levy et al. (2012)*. 96 strains were grown from glycerol stocks in a 96-well plate containing Synthetic Complete medium (0.67% Yeast Nitrogen Base without amino acids and 0.079% Complete Synthetic Supplement (ForMedium, UK)) with 2% glucose (SCD) for 24 hr at 30°C. The cells were diluted 1:50 in fresh medium, grown for 20 hr and diluted again 1:50 in fresh medium. Finally, cells were grown for 4 hr, cell densities were determined by OD at 600 nm in a Tecan plate reader and then were diluted to another plate containing SCD or appropriate medium required for microscopy experiment using a Biomek NX (Beckman Coulter) liquid handling robot, capable of pipetting variable volumes of cells across wells in a 96 plate, to a target density of ~17000 cells/µl. This minimized any possible bias due to variability in cell densities among strains. A final 5-fold dilution was done by pipetting 80 µl cells onto a pre-coated 96-well microscopy plate containing 320 µl of SCD. The microscopy plate was then sealed with LightCycler 480 sealing foils (Roche), cells were spun at 450 rpm for 2 min and taken for microscopy observations.

Microscopy plates (96-well glass bottom, MGB096-1-2-LG-L, Brooks Life Science Systems) were coated with 200 µl sterile solution of 200 µg/ml concanavalin A (type IV, Sigma) at 37°C for 16–18 hr. The solutions were then pipetted out and the plates were washed twice with sterile milli-q water. Plates were dried at 4°C for at least 24 hr. Imaging was performed using an ImageXpress Micro (Molecular Devices) microscope, with laser autofocusing, at an interval of 90 min for up to 12 hr. The microscope chamber was maintained at 30°C.

### Image processing

Images were processed using custom scripts written in perl. Yeast cells were identified by juxtaposition of bright and dark pixels (10,36). A pixel was considered 'bright' if its intensity exceeded mean +2.2 s.d. value and a pixel was considered 'dark' if its intensity was below mean-2.2 s.d. value. In addition, Sobel's edge detection algorithm (*Sobel and Feldman, 1968*) was applied for identifying yeast cell boundaries with sharp changes in pixel intensity. Clustering was used to identify the microcolonies and the centroid position for each microcolony was calculated. Microcolonies were tracked through centroid tracking over time. Sudden increase or decrease in centroid number in a time series indicated a failure in image acquisition or image processing and such images were discarded from the analysis. To differentiate cells from cellular debris, residuals of concanavalin A coating etc., two filtering steps were used. First, only objects that were bigger than 50 pixels at the start of observation were considered. Second, the object had to increase its size to greater than 2-fold at the end of observation. Whether neighbouring colonies touch each other at any point in time during

microscopy observation was also checked. If they did, they were tracked only up to the time they touched each other. Similar to the findings of Ziv et al. (*Ziv et al., 2013*), lag phase during growth was observed in some microcolonies in our experiments. In addition, growth slowdown near the end of observation for some microcolonies was also observed, possibly due to nutrient limitation (*Figure 5—figure supplement 4*).

To calculate growth rate, linear regression on natural log-transformed area vs. time for three consecutive time points was performed and only fits with $R^2 \geq 0.9$ were considered. This was repeated using a three-point moving window over all time points. Of all these regressions, the maximum value was chosen as the microcolony growth rate to avoid biases because of slow down of growth during lag and/or due to possible substrate limitation near the end of observation.

## Screening of deletion mutants, classification and functional enrichment analysis

Growth distributions for deletion mutants were measured in three independent experiments on different days. To calculate reproducibility of growth rate between replicates, mean growth rate was allowed to vary up to 0.05 h$^{-1}$ and then Kolmogorov-Smirnov distance (K-S distance) (*Justel et al., 1997*) was calculated between all replicates after shifting one of them (through addition/subtraction) by difference in mean growth rates. Three replicates were considered as three nodes in a graph with K-S distance between them as the edge weight. If the K-S distance between two replicates exceeded 0.1, no edge was drawn between those two nodes. The sub-graph where the maximum number of nodes was connected to each other directly and via shortest possible distance was considered as reproducible replicates. Only mutants with at least two reproducible replicates were considered in our analysis. The number of reproducible replicates for each mutant is given in *Supplementary file 2*. The proliferation distributions for all mutants are shown in *Supplementary file 4*.

To calculate slow fraction from a proliferation distribution, first, a cumulative distribution function (cdf) was calculated with density being calculated at an interval of 0.01 h$^{-1}$. The cdf function was then scanned for maximum slope using a window of 5 points. At the point with maximum slope, a line with the maximum slope was fitted and the points that deviated from the fitted line by >0.02 h$^{-1}$ were considered as the edges of the main subpopulation. In the next step, if the left sub-population was bigger than the right sub-population, the percentage of slow fraction was calculated as (% left sub-population-% right sub-population) and the percentage of fast sub-population was set to zero. If the right sub-population was bigger than the left sub-population, the percentage of fast fraction was calculated as (% right sub-population-% left sub-population) and the percentage of slow sub-population was set to zero.

If the mode of a growth distribution is reduced (compared to WT) and the growth rate of the slow fraction is not reduced, the main sub-population growth distribution is likely to overlap with and mask a slow growing sub-population. To avoid such scenarios, all the reproducible growth distributions for the WT strain were collected and the mode growth rate was computationally reduced in steps of 0.01 h$^{-1}$ without changing the growth rate of the slow sub-population. The percentage slow fraction was calculated at each step. As expected, reduction in mode growth rate without moving the slow fraction led to a reduction in % of slow fraction (*Figure 2—figure supplement 2A*).

Mutants with altered mode proliferation rate compared to WT strain were identified through Mann-Whitney U test (FDR < 0.1). Mutants with altered slow fraction were identified by Mann-Whitney U test (FDR < 0.1) after correcting for any change in mode growth rate (*Figure 2—figure supplement 2A*). Comparison between replicate measurements of mode growth rate and percentage of slow fraction was done (*Figure 5—figure supplement 5A*). The mean proliferation rate of mutant strains obtained in our assay was comparable with published values (*Figure 5—figure supplement 5B*).

We used GOslim gene annotation (*Gene Ontology Consortium, 2015*; *Gene Ontology Consortium, 2018*) for functional class enrichment analysis and we performed a hypergeometric test as follows. Let us assume that in a group 'g' from screening (for example, the group with increased slow fraction but no change in mode growth rate), out of total $N_g$ genes, $X_g$ genes are associated with function $f$ according to GOslim annotation. Let us also assume that out of total N genes screened in our data, X genes belong to the functional class $f$ according to GOslim annotation. Thus, the probability that the group 'g' contains more number of genes of functional class $f$ than expected by

chance alone is given by $\mathrm{p} = \sum\limits_{i=X_g}^{N_g} \dfrac{\dbinom{X}{i}\dbinom{N-X}{N_g-i}}{\dbinom{N}{N_g}}$, which gives the p-value. A further multiple testing

correction was done using Benjamini-Hochberg procedure with FDR<0.1.

## Quantification of incomplete penetrance

Incomplete penetrance was calculated for all mutants that showed significant reduction in mean pro-liferation rate compared to the WT strain (Mann-Whitney U test, FDR < 0.1). For each of these mutants, replicate proliferation distributions were compared with replicate proliferation distributions of WT strain that were reproducible across the screening experiment. Average overlap of the mutant proliferation distributions with the bulk sub-population of each of the WT proliferation distribution was calculated. For WT strain, to calculate bulk sub-population from a proliferation distribution, first, a cumulative distribution function (cdf) was calculated with density being calculated at an interval of 0.01 h$^{-1}$. The cdf function was then scanned for maximum slope using a window of 5 points. At the point with maximum slope, a line with the maximum slope was fitted and the point that deviated from the fitted line by >−0.02 h$^{-1}$ was considered as the edge of the bulk sub-population. Thus, for each mutant, this led to a distribution of a percentage of cells showing WT-like proliferation (*Figure 2—figure supplement 1*). In the next step, it was tested whether the distribution of percentage of WT-like cells was significantly different from zero (Wilcoxon rank-sum test for one sample) and an FDR correction for multiple testing was performed (FDR < 0.1).

## Mitotracker green and TMRE staining

To perform mitotracker green (MitoTracker Green FM, Molecular Probes, Thermo Fisher Scientific) staining, cells were centrifuged at maximum speed for 2 min and washed twice with buffer contain-ing 10 mM HEPES (pH 7.4) and 5% glucose. Cells were then re-suspended in the same buffer and Mitotracker Green (10 μM stock dissolved in DMSO) was added to a final conc. of 100 nM. Cells were incubated for 20 mins at 30°C, washed twice with PBS (pH 7.4) and quantified by flow cytome-try (LSR Fortessa, BD Biosciences).

TMRE (Tetramethylrhodamine, Ethyl Ester, Perchlorate) (Molecular Probes, Thermo Fisher Scien-tific) is a positively charged dye that accumulates inside mitochondria depending on the mitochon-dria transmembrane potential generated due to transfer of protons across mitochondrial membrane resulting in net negative charge inside the mitochondria (*Crowley et al., 2016*). For TMRE staining, cells were grown as in pre-growth step in microcolony assay, precipitated, washed twice with PBS, were re-suspended in PBS, and TMRE was added to a final conc. of 100 nM from a 10 mM stock dis-solved in DMSO. Cells were incubated at 30°C for 30 min, were washed twice with PBS and were analysed by flow cytometry or were sorted. There was a gap of 15–20 min between the end of stain-ing and beginning of flow cytometry experiments due to the time required for cleaning, priming and setting up of flow cytometry machine parameters. Day-to-day variations were observed in measure-ment of TMRE distributions.

## Cell sorting and growth measurement of sorted bins

Cells were sorted by TMRE signal into four bins HI, M1, M2, LO (*Supplementary file 5*, *Figure 3C*) in an Aria II SORP cell sorter (BD Biosciences). For growth rate measurement, stress resistance mea-surement, and mitochondrial DNA quantitation by qPCR in the sorted bins, 100,000 cells per bin were sorted at room temperature into 1.5 ml tubes pre-filled with 600 μl of PBS. After sorting, 200 μl of YPD was added to each tube, cells were centrifuged for 5 min at maximum speed at room tem-perature, and the supernatant was thrown away. Cells were re-suspended in 600 μl of PBS before proceeding for subsequent experiments. For heat shock experiments, 100 μl of sorted cells were put into PCR tubes and were subjected to heat shock in a PCR machine, put on ice for 1 min before measurement of growth and viability. For RNA sequencing experiments, 750,000 cells per bin were sorted in three 1.5 ml tubes, each pre-filled with 800 μl of PBS. After sorting, 200 μl of YPD was added to each tube, centrifuged at maximum speed for 5 min, and supernatants were discarded. The cell pellets were gently washed twice using PBS. Total RNA was isolated using MasterPure yeast RNA isolation kit (Epicentre) following manufacturer's protocol. Cells from four sorted bins were

grown in SCD medium up to 48 hr and their growth distributions were remeasured using high-throughput microscopy assay (*Figure 5—figure supplement 6A*).

To determine percentage of respiration deficient cells, cells were plated on plates containing Synthetic Complete medium with 3% glycerol and 0.1% glucose (SCDG) solidified with 1.5% agar to a target density of ~100–150 colonies per plate. After 5–7 days, number of small and big colonies were counted and the percentage of respiration deficient cells were determined as - percentage of respiration deficient cells = $\frac{\text{Number of small colonies}}{\text{(Number of small+big colonies)}} \times 100$

Respiration deficient cells showed almost no mtDNA and remained slow growing even after seven days of growth in Synthetic Complete medium with 2% glucose (SCD) (*Figure 5—figure supplement 6B,C*).

Cells were also tested for switching of respiration capability (*Figure 5—figure supplement 6D, E*). Equal number of sorted cells from each bin was plated onto SCDG plates (with 0.1% glucose and 3% glycerol as the carbon sources) and SCG plates (with 3% glycerol as the carbon source) and allowed to grow for 5 days. Percentage of cells regaining respiration capability was calculated as (No. of respiration capable cells on SCDG plate – No. of colonies on SCG plate)/Total no. of colonies on SCDG plate.

## Growth rate switching in microcolonies

To test whether microcolonies switch from fast to slow or slow to fast growth rate, microcolony growth rates at all time points of tracking were calculated. Growth rate of a microcolony at a time point was calculated using linear regression of ln(area) with time including one preceding time point and one subsequent time point and only fits with $R^2 \geq 0.9$ were considered. In case of two cutoff values (say 'c1' as the lower cutoff and 'c2' as the higher cutoff) for growth rate for determining slow and fast growing microcolonies, if a microcolony showed growth rate above 'c2' for at least three consecutive time points and afterwards showed growth rate below 'c1' in at least three consecutive time points and the time of switching from fast to slow growth was not within last three time points of observation (to avoid incorrect classification due to slowdown in growth at later time points), the colony was classified to be switching from fast to slow growth (*Figure 5—figure supplements 7, 8* and *9*). Similarly, if a microcolony showed growth rate below 'c1' for at least three consecutive time points and afterwards showed growth rate above 'c2' in at least three consecutive time points and the time of switching from slow to fast growth was not within first 3 time points of observation (to avoid incorrect classification due to the lag phase), the colony was classified to be switching from slow to fast growth (*Figure 5—figure supplements 7, 8* and *9*). In case of a single cutoff value, the same criteria were applied as above with 'c1' being equal to 'c2'. This was applied to identify switching in cells of TMRE sorted bins HI, M1, M2 and LO in the WT strain. Various cut-off values for identifying switching were tested to check switching of microcolony growth rates across different ranges of growth rates (*Figure 5—figure supplements 7, 8* and *9*).

For calculating switching in unsorted WT and deletion strains, the criteria for classification were slightly modified since there were fewer time points of observation. Specifically, if a microcolony showed growth rate above 'c2' for at least two consecutive time points and afterwards showed growth rate below 'c1' in at least two consecutive time points and the time of switching from fast to slow growth was not within last 2 time points of observation (to avoid incorrect classification due to slowdown in growth at later time points), the colony was classified to be switching from fast to slow growth (*Figure 5—figure supplement 7*). Similarly, if a microcolony showed growth rate below 'c1' for at least two consecutive time points and afterwards showed growth rate above 'c2' in at least two consecutive time points and the time of switching from slow to fast growth was not within first 2 time points of observation (to avoid incorrect classification due to lag), the colony was classified to be switching from slow to fast growth (*Figure 5—figure supplement 7*).

## Switching of cells from high to low membrane potential

To test for switching between high and low TMRE states, cells with high, medium or low TMRE were grown for 48 hr after sorting and their mitochondrial membrane potential values were remeasured (*Figure 5—figure supplement 10*). Cells from the HI TMRE bin consisted of 99% of high TMRE cells and 1% low TMRE cells (impurity). As measured by time-lapse microscopy (*Figure 5—figure supplement 11*), high TMRE cells in the HI bin consisted of slow-growing (80%) and fast growing (20%)

cells. Low TMRE cells in the HI bin (impurity) were assumed to be fast growing, as this gave a conservative estimate for the number of cells switching from high to low TMRE state in 24 hr. The mean growth rates of the slow and fast-growing cells (geometric mean) were 0.20 $h^{-1}$ and 0.37 $h^{-1}$ respectively (*Figure 5—figure supplement 11*). The percentages of cells switching from low TMRE to high TMRE state in 24 hr was estimated as the average switching rate of cells from M1, M2 and LO bins to high TMRE state and it was estimated to be ~10% in 24 hr (*Figure 5—figure supplement 10*). Given these data, assuming exponential growth for all sub-populations over 24 hr and in the absence of switching from high to low TMRE state,

$$\% \,\text{of low TMRE cells} = \frac{0.01 \times e^{0.37 \times 24} - 0.1 \times \left(0.01 \times e^{0.37 \times 24}\right)}{0.80 \times e^{0.20 \times 24} + 0.19 \times e^{0.37 \times 24} + 0.01 \times e^{0.37 \times 24}} \times 100\%$$

$$= \frac{71.87 - 7.19}{97.20 + 1365.49 + 71.87} \times 100\% = 4.21\%$$

This estimate is 8-fold lower than the observed % of low TMRE cells (34.7%).

For growth of cells from 24 hr to 48 hr, assuming exponential growth for 24 hr and no switching from high to low TMRE state,

$$\% \,\text{of low TMRE cells after 48 hr} = \frac{0.347 \times e^{0.37 \times 24} - 0.1 \times \left(0.347 \times e^{0.37 \times 24}\right)}{0.80 \times 0.653 \times e^{0.20 \times 24} + 0.20 \times 0.653 \times e^{0.37 \times 24} + 0.347 \times e^{0.37 \times 24}} \times 100\%$$

$$= \frac{2493.81 - 249.4}{63.47 + 1437.36 + 2493.81} \times 100\% = 56.2\%$$

Again, this estimate is substantially lower than the observed % of low TMRE cells (76.5%). Taken together, these results suggest considerable switching from high to low TMRE state.

## Measurement of mtDNA copy number by qPCR

To determine mtDNA copy number per cell using quantitative PCR (qPCR), five primer pairs specific to nuclear DNA (ACT1, ALG9, KRE11, TAF10, COX9) and five primer pairs specific to mitochondrial DNA (COX1, ATP6, COX3, ATP9, tRNA – primer picked around tQ(UUG)Q gene) were used (*Supplementary file 3*). A standard curve for each of primer was made, using six concentrations of genomic DNA serially diluted from the highest concentration by 4-fold at each step. Absolute quantification of DNA copy number was performed using the standard curve. Three technical replicates for each primer and for each sample were set up totaling 30 reactions per sample. To compare mtDNA copy number across sorted bins, nuclear DNA and mtDNA copy numbers in all bins were normalized by the respective values for LO bin. Two sample t-test was used to check whether the normalized value for nuclear DNA differs significantly from the normalized value for mtDNA and a p-value was calculated using a two-sample t-test. Mean mtDNA copy number per cell was calculated by the ratio of mtDNA to nuclear DNA and standard deviations were calculated by taking error propagation models into account.

To overexpress the MIP1 gene, the MIP1 gene under the control of the native promoter (930 bp upstream and 262 bp downstream, total insert length - 4957 bp) was cloned into pRS413 plasmid and then transformed into NEB 10B electrocompetent *E. coli* cells. The plasmid with the verified construct was then isolated and transformed into yeast cells.

## RNA sequencing experiment and data analysis

Isolated total RNA (using MasterPure yeast RNA isolation kit (Epicentre)) was checked and quantified using bioanalyzer. 200 ng of total RNA for each sample was taken and was mixed with 4 µl of 1:1000 dilution of ERCC spike-in mix1 (Thermo Fisher Scientific). Sequencing was done in Illumina HiSeq with paired end 2 × 50 bp reads. Quality of the sequenced reads was checked using FastQC (*Andrews, 2016*) and then the reads were mapped to reference yeast transcriptome (R64-1-1 reference cdna sequence from Ensembl [*Hubbard et al., 2002*]) using bowtie2 (*Langmead and Salzberg, 2012*). Mapping statistics was calculated using a custom script where only read pairs mapping concordantly and uniquely to the reference sequence were considered. The data were normalized using ERCC spike-in reads as controls using RUVg method in R package RUVSeq (*Risso et al., 2014*). Correlation between replicates were checked through distance heatmap and PCA analysis (*Figure 5—figure supplement 12A,B*), using R package DESeq2 (*Love et al., 2014*). Differentially expressed genes were identified using package DESeq2. Functional enrichment analysis on sets of

differentially expressed genes was done using a hypergeometric test as described above with multiple testing correction (FDR < 0.1) (Benjamini-Hochberg method) with GOslim gene annotations.

### Reconstruction of single mutants

Gene deletion mutants were remade in the WT strain using sequence specific homologous recombination. First, the deletion cassette from the appropriate deletion strain from the collection was amplified using primers such that the amplified region contained the deletion cassette with KanMX marker and 50–300 bp of overhang on either side of the cassette. Particular care was taken to avoid neighbouring genes from being amplified. The PCR product was transformed into competent yeast cells (prepared using lithium acetate and PLI – made by mixing 1 ml water, 1 ml 1M lithium acetate and 8 ml 50% PEG3350) and colonies were selected on G418 plates. Two verified clones for each mutant were picked for experiments. Some of the mutants associated with mitochondrial function were found to be compensated in the deletion collection (*Figure 5—figure supplement 12C,D*). Beyond the initial high-throughput measurement of proliferation distribution of deletion mutants, all experiments were performed with freshly made deletion mutants.

### Long-term microscopy-based growth measurements for measurement of stress tolerance and drug resistance

To observe growth of yeast cells in drug (fluconazole dissolved in DMSO, stock conc. 5 mg/ml) over 7 days, yeast cells were imaged under the microscope every ~24 hr. To have a bigger part of a well imaged and to increase the number of data points, 48 fields of view per well were imaged in these experiments. Yeast cells and microcolonies were identified as above. Before tracking the microcolonies over time, the images for all fields of view in a well were merged which allowed tracking of microcolonies even if the plate was positioned slightly differently in the microscope at different time points. Microcolonies were tracked over time and growth rates were calculated. A growth rate of $0.02$ h$^{-1}$ after the first time point was taken as cut-off for survival on fluconazole, as most colonies showed initial growth but then stopped growing. Percentage survival in heat shock and hydrogen peroxide treatment was calculated as the ratio of the number of colonies showing growth under stressed condition compared to the total number of colonies showing growth under unstressed condition.

### Measurement of respiration capability in drug resistant cells

To test whether the cells that survive fluconazole treatment can still respire, the drug resistance of the sorted sub-populations from the HI and LO bins were measured on agar plates after 15 days of growth in SCD medium supplemented with fluconazole (9.5 or 10 µg/ml) and solidified with 1.5% agar. This assay needed lower concentrations of fluconazole compared to the microscopy-based assay, as only the colonies that divided multiple times were visible on the plate. Sorted cells from bins HI and LO were plated directly after sorting onto the drug plates (5–6 replicates per bin), onto plates without any drug as well as onto SCDG plates to calculate the percentage of cells capable of respiration. Cells were counted after 15 days and 40–50 colonies from each plate were randomly picked and checked for respiration capability by plating onto plates containing 3% glycerol as the carbon source.

### Data availability

RNA-sequencing data that support the findings of this study have been deposited in NCBI GEO with the accession code GSE104343. Microscopy images have been submitted to openmicroscopy. org. The raw microcolony growth data for the WT and mutant strains are available at https://github.com/lehner-lab/MicroscopyCode-Dhar_et_al/tree/master/Microscopy_screen_processed_data.

### Code availability

Custom codes for analysing microscopy images are available at https://github.com/lehner-lab/MicroscopyCode-Dhar_et_al (*Dhar and Faure, 2019*; copy archived at https://github.com/elifesciences-publications/MicroscopyCode-Dhar_et_al).

## Acknowledgements

Work in the lab of BL was supported by a European Research Council Consolidator grant (616434), the Spanish Ministry of Economy and Competitiveness (BFU2017-89488-P and SEV-2012–0208), the AXA Research Fund, the Bettencourt Schueller Foundation, Agència de Gestió d'Ajuts Universitaris i de Recerca (AGAUR SGR 1322), the EMBL Partnership, and the Generalitat/CERCA program. Work in the lab of LBC was supported by MINECO (BFU2015-68351-P) and AGAUR (2014 SGR 0974) and the Unidad de Excelencia Maria de Maeztu (MDM-2014–0370). The authors thank Dr. Raul Gomez Riera and the CRG Advanced Light Microscopy Unit for assistance with high-throughput microscopy, the CRG/UPF FACS facility for help with flow cytometry, the CRG Genomics Unit for RNA-seq and DNA sequencing experiments, and the UPF Genomics facility for help with automated liquid handling. RD was partially supported by the Swiss National Science Foundation Early Postdoc Mobility fellowship.

## Additional information

### Funding

| Funder | Grant reference number | Author |
|---|---|---|
| H2020 European Research Council | 616434 | Ben Lehner |
| AXA Research Fund | | Ben Lehner |
| Ministerio de Economía y Competitividad | BFU2011-26206 | Ben Lehner |
| Fondation Bettencourt Schueller | | Ben Lehner |
| Ministerio de Economía y Competitividad | BFU2015-68351-P) | Lucas B Carey |
| Agència de Gestió d'Ajuts Universitaris i de Recerca | | Ben Lehner Lucas B Carey |
| Schweizerischer Nationalfonds zur Förderung der Wissenschaftlichen Forschung | | Riddhiman Dhar |

The funders had no role in study design, data collection and interpretation, or the decision to submit the work for publication.

### Author contributions

Riddhiman Dhar, Conceptualization, Data curation, Software, Formal analysis, Validation, Investigation, Visualization, Methodology, Writing—original draft, Writing—review and editing, set up high-throughput microcopy assays, developed pipelines for image processing, performed high-throughput microscopic proliferation measurements of the deletion mutants and analyzed the data, performed the remaining experiments, analyzed the data, and wrote the manuscript; Alsu M Missarova, Conceptualization, Formal analysis, Investigation, Writing—review and editing, performed high-throughput microscopic proliferation measurements of the deletion mutants, analyzed the data and contributed to writing the manuscript; Ben Lehner, Lucas B Carey, Conceptualization, Supervision, Funding acquisition, Writing—original draft, Writing—review and editing

### Author ORCIDs

Riddhiman Dhar http://orcid.org/0000-0003-4642-0492
Alsu M Missarova http://orcid.org/0000-0001-9472-2095
Ben Lehner http://orcid.org/0000-0002-8817-1124
Lucas B Carey https://orcid.org/0000-0002-7245-6379

### Decision letter and Author response

Decision letter https://doi.org/10.7554/eLife.38904.042

Author response https://doi.org/10.7554/eLife.38904.043

# Additional files

## Supplementary files

• Supplementary file 1. Mean and Mode growth rate ($h^{-1}$) and % slow fraction for the natural yeast strains from SGRP collection.
DOI: https://doi.org/10.7554/eLife.38904.029

• Supplementary file 2. Mean, median and mode growth rates ($h^{-1}$), Standard deviation (SD), Noise (Coefficient of variation, CV), % slow fraction, number of replicates showing reproducible results and the classification colour code (as in *Figure 2A*) for all the mutants with reproducible results.
DOI: https://doi.org/10.7554/eLife.38904.030

• Supplementary file 3. Primer pairs used for quantifying mtDNA copy number using quantitative PCR.
DOI: https://doi.org/10.7554/eLife.38904.031

• Supplementary file 4. Proliferation distributions of 1520 deletion mutants for which reproducible measurements were obtained. Multiple lines in each plot represent reproducible replicate measurements. x-axis represents microcolony growth rate ($h^{-1}$) and y-axis represents density.
DOI: https://doi.org/10.7554/eLife.38904.032

• Supplementary file 5. An example of gating strategy used for cell sorting experiments.
DOI: https://doi.org/10.7554/eLife.38904.033

• Supplementary file 6. Key Resources Table.
DOI: https://doi.org/10.7554/eLife.38904.034

• Transparent reporting form
DOI: https://doi.org/10.7554/eLife.38904.035

## Data availability

RNA-sequencing data that support the findings of this study have been deposited in NCBI GEO with the accession code GSE104343. Microscopy images are available via the Image Data Resource repository under accession number S-BIAD2. The raw microcolony growth data for the WT and mutant strains are available at https://github.com/lehner-lab/MicroscopyCode-Dhar_et_al/tree/master/Microscopy_screen_processed_data.

The following datasets were generated:

| Author(s) | Year | Dataset title | Dataset URL | Database and Identifier |
|---|---|---|---|---|
| Dhar R, Missarova AM, Lehner B | 2018 | Single cell functional genomics reveals the importance of mitochondria in cell-to-cell phenotypic variation | https://www.ncbi.nlm.nih.gov/geo/query/acc.cgi?acc=GSE104343 | Gene Expression Omnibus, GSE104343 |
| Riddhiman Dhar, Alsu M Missarova, Ben Lehner, Lucas B Carey | 2019 | Microscopy image data from: Single cell functional genomics reveals the importance of mitochondria in cell-to-cell phenotypic variation | https://www.ebi.ac.uk/biostudies/studies/S-BIAD2 | EMBL-EBI BioStudies, S-BIAD2 |

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
