## [Decision Letter]

Thank you for submitting your article "Single cell functional genomics reveals the importance of mitochondria in cell-to-cell phenotypic variation" for consideration by *eLife*. Your article has been reviewed by three peer reviewers, and the evaluation has been overseen by Naama Barkai as the Senior and Reviewing Editor. The reviewers have opted to remain anonymous.

Please address all comments in the individual reviews below. In particular, the reviewers were concerned that the experiments showing transition from 'slow' to 'fast' state are important but are not very convincing and should be addressed. in this context, please note the paper from Gilles Charvin (Fehrmann et al., 2013). They used microfluidics to look at mitochondrial membrane potential in individual yeast cells as they aged. This paper should have been cited in the current manuscript, and it sets the bar both technically (for the microfluidic analysis of membrane potential and growth in individual cells) and conceptually (they found that decline of membrane potential is age-independent and leads to clonal senescence – i.e., is not reversible). The phenomenon studied in the current manuscript is different because it has to do with high-potential/low-respiration cells, but the Charvin study sets the precedent in the literature at least for irreversible mitochondrial-state changes.

Reviewer #1:

Dhar et al. address an important topic of broad interest: cellular heterogeneity and its relationship to proliferation and survival. Given that very little is known about the causes of heterogeneity, their approach of doing a genetic screen for yeast mutants that alter the distribution of growth rates within a population is sensible, and the scale of the experiment (>1500 mutants) is impressive. However, despite my enthusiasm for the research question and the general approach, I do have some major concerns:

1) The authors appear to have reinvented the micro-colony growth assay of Levy et al. (Levy et al., 2010) and should do a better job of presenting the similarities and differences between the two methods. Of particular concern is that they appear to have cell behaviors that were not seen by the other group – they mention lag (subsection “Image processing” and subsection “Cell sorting and growth measurement of sorted bins”, end of last paragraph) and nutrient deprivation, neither of which was seen by the other group (Levy et al., 2010 and Ziv et al., 2013) in medium with sufficient glucose.

2) The finding of a role for mitochondria in growth heterogeneity is interesting, but petites are well known to decrease growth rate and increase stress resistance. So the key advance here would be the finding that mitochondrial states switch back and forth, rather than petites just being a slow-growing dead end. Levy et al., 2010, had already demonstrated switching of growth states, and that single micro-colonies could give rise to the complete distribution of growth rates. So the question is whether the switching is driven by mitochondrial-state changes. Unfortunately, the experiments on switching are not entirely convincing. The authors took bins of cells sorted by TMRE (mitochondrial membrane potential) and assayed their growth. They found that high-TMRE cells tended to have slower growth rates and that high TMRE correlated with respiration deficiency. These results would be consistent with switching only in the direction of becoming petite. But the authors argue that switching can go in both directions because sorted bins could repopulate the full distributions of TMRE staining and growth. However, noise in TMRE staining or other technical reasons could mean that the sorted bins were contaminated to some extent with respiration-proficient cells, which then might be responsible for the repopulation of the distributions. A more direct tracking experiment is necessary.

3) The manuscript does not offer much mechanistic insight into the changes in mitochondrial state or mtDNA copy number. What is causing these changes? Why do the changes persist for the number of cell generations that they do? Is the mitochondrial-state phenomenon tied to, or independent from, the heterogeneity in gene expression seen in Levy et al., 2010?

Reviewer #2:

In this manuscript, Dhar et al. studied heterogeneity of growth between yeast cells by measuring single-colony growth rates of a large collection of yeast strains (gene deletions and natural strains). The originality was to track over time the size of colonies that emerge from single cells, deriving individual growth rates of these initial founder cells. The distribution of single-cell growth rates differed between strains and the authors found marked alterations of these distribution for mutants of nuclear genes involved in mitochondrial functions. They sorted cells based on TMRE staining, and observed that high signal correlated with i) slow growth ii) low mtDNA iii) low competence for respiration iv) peculiar transcriptomic signatures v) sensitivity to heat and oxidative stress but vi) higher survival to fluconazole. Overexpression of MIP1 increased mtDNA content and reduced both the fraction of slow growing cells and the fraction of respiratory-deficient cells.

General Assessment:

This is a very sound study. The amount of work is impressive and the complementarity of the experimental approaches is remarkable. Conclusions that heterogeneity in mitochondrial 'state' affects phenotypic heterogeneity between cells is very important and opens novel avenues for investigating cell-cell biological variation, which is a hot topic.

1) The raw data of colony growth assays is not made available. This is not acceptable, as others may later want to explore the dataset in various ways. The raw data should be uploaded in a public repository in two forms:

- Time-course data of the size of each colony, for all colonies and all yeast strains. This is important if others want to explore time-series for lag phase, saturations or other traits that may not be fully correlated to the growth rates computed by the authors.

- The scalar values of growth-rates of individual colonies, so that others can use these values directly.

2) GO analysis (Figure 2E) highlighted respiration based on many genes but little enrichment. Other terms were better enriched, although among fewer genes, especially for the 'red' set. The authors should mention in text what are the kinases, cytoskeletal binding proteins and carbohydrate transporters involved in maintaining the% slow fraction.

Reviewer #3:

The described research aims to identify the underlying mechanisms of investigating proliferation heterogeneity in isogenic cell populations. Through a quantitative microcolony assay, the authors demonstrate in isogenic cultures of yeast maintain heterogenous and stable rates of cell division that are influenced by genetic variation. By screening the yeast deletion collection, the authors discover that mitochondrial activity influences proliferation heterogeneity, and that mitochondrial membrane potential and respiratory dysfunction are features of the "slow fraction" of these cell populations. This work highlights the importance of mitochondrial dynamics as a master regulator of cells and how these dynamics may influence heterogeneity.

• *S. cerevisiae* is known to throw off petite cells that lack complete mtDNAs, in a genetic background dependent manner (these petites can also propagate in liquid cultures so multiple serial transfers can alter the initial number of petites in the inoculums which may influence the overall observed heterogeneity differences between strains). The "slow fraction" could represent these non-respiring petite cells, however the authors provide evidence that the "fast" to "slow" transitions are reversible (i.e. that the High TMRE, respiratory deficient cells could give rise to Low TMRE cells (subsection “Mitochondria state but not content predicts slow growth”, last paragraph, Figure 5—figure supplement 2Figure). Have these "switched" cells regained respiratory capability? To decide if cells have switched between slow and fast growth, the authors rely on a threshold value of growth rates (as opposed to a percent change). It's not clear why the 0.3 h-1 threshold was chosen. As I understood, a microcolony that changes from 0.1 to 0.29 h-1 would be scored as slow growing while a microcolony that changes from 0.29 to 0.31 h-1 would be scored as one that switched between slow and fast growth. Can the authors clarify?

• Yeast cells growing in microplates are likely to be oxygen starved (even if being shaken) and near transition states in metabolism, so it makes sense that changes in mitochondrial "states" are being observed. A discussion of how environmental conditions could influence heterogeneity would help to put this into context.

[Editors' note: further revisions were requested prior to acceptance, as described below.]

Thank you for submitting your article "Single cell functional genomics reveals the importance of mitochondria in cell-to-cell phenotypic variation" for consideration by *eLife*. Your revised article has been reviewed by two peer reviewers, and the evaluation has been overseen by Naama Barkai as the Senior and Reviewing Editor. The reviewers have opted to remain anonymous.

The reviewers remained supportive of your study and appreciated most of the revisions. However, the evidences supporting the claim that cells switch from a low-growth to a fast-growing mode remained inconclusive. Since this is not the essence of your results, we are happy to accept the paper. however, we ask that you will back off this claim and move Figure 5 to the supplementary material.

The reviewers also felt that the discussion of previous findings, and their relation to your work, will improve the manuscript. please try to do that as well. see below specific comments.

Reviewer #1:

In this revised manuscript, Dhar et al. present new analyses aiming to address concerns about their inferences of switching from a slow-growing, respiration-deficient cell state to a fast-growing, respiration-capable cell state. I remain enthusiastic about the topic of the manuscript and I remain impressed by the scale of the mutant analysis, but I also remain concerned that the authors' analyses of switching rely on indirect arguments that are not entirely convincing. Specifically:

1) They argue (subsection “Switching of cells from high to low membrane potential”) that TMRE-high to TMRE-low switching must take place to explain the observation that 1% of cells are TMRE-low immediately after sorting into the TMRE-high bin (a low rate of contamination), whereas 34.7% of cells are TMRE-low 24 hours later. Their argument relies on a simple model in which TMRE-high cells consist of two subpopulations (80% slow-growing and 20% fast-growing), TMRE-low cells are fast-growing, and the average slow and fast growth rates represent the corresponding subpopulations. This model has some problems. First, the arithmetic mean growth rate might not be appropriate to model changes in subpopulation sizes over time. If each cell chooses a growth rate at random from its distribution then what matters is the geometric mean, which is heavily influenced by low values. Alternatively, if growth rate is relatively stably inherited through time then what matters is the fastest members of a subpopulation. Either way, arithmetic mean could be a very poor indicator of subpopulation growth. Second, even if we ignore the problem with the arithmetic mean and take the authors' argument and apply it to the transition between the 24-hr time point and the 48-hr time point, then their calculation actually predicts quite well the 48-hr subpopulation sizes from the 24-hr starting point, so TMRE-high to TMRE-low switching is not required to explain this transition. Third, the authors' simple model ignores lag, which would likely further under-represent the slow-growing (high-TMRE) fraction, and might actually be why the transition from 0 to 24 hours requires additional explanation whereas the transition from 24 to 48 hours does not.

2) The authors have verified that their analyses of growth rate switching in micro-colonies are robust to changes in cutoffs between "fast" and "slow" growth, which is good, but what these switches mean is insufficiently discussed. In particular, fast-to-slow switches such as those shown in Figure 5D necessarily involve concerted changes in most cells in a micro-colony because a single cell changing from fast to slow in the middle of micro-colony growth would not appreciably change the growth rate of the micro-colony. The authors should discuss (and perhaps experimentally investigate) what such switches at the level of the micro-colony mean.

3) The conclusion that some cells regain ability to respire is an interesting piece of this work but it is not clear exactly what the authors mean by it. In Figure 5E they present the "percentage of colonies that regained capability to respire" but the calculation involves two different plates that were not replicas, so the ability to "regain" respiration was a more indirect inference than the text suggests.

Because of the above concerns, my view remains that more direct evidence of switching at the level of single cells (from high-TMRE to low-TMRE, from slow growth to fast growth, and from respiration-deficient to respiration capable) would substantially strengthen this paper.

Reviewer #3:

I completely agree that the resubmission only indirectly addresses our concerns about slow to fast growth. Even with the new statistical analysis, some quick growing contaminants could explain what they conclude are transitions. A simple replica plating experiment would have shown that their non-respiring cells can convert back to respiring cells. Given that they were previously asked for alternate experimentation to demonstrate the slow to fast transition, they need to back off this claim. (and minimize and move Figure 5 to supplemental).

In general, the authors could do better at discussing their findings in the context of previously published work.

Their proposal (in the Discussion) that petites are part of a mitochondrial heterogeneity continuum is not substantiated if they are unable to convincingly show reversibility.

In the Abstract, they should change the word "developed" a high-throughput method to "used".

The authors now refer to mitochondrial membrane potential in places though there are still several places where mitochondrial "state" is used and could/should be corrected (e.g. Subsection “Mitochondrial membrane potential but not amount predicts slow growth”, fourth and fifth paragraphs, subsection “Variation in mitochondrial state predicts additional phenotypic heterogeneity including drug resistance”, first paragraph and especially in the first paragraph of the Discussion).

---

## [Author Response]

Please address all comments in the individual reviews below. In particular, the reviewers were concerned that the experiments showing transition from 'slow' to 'fast' state are important but are not very convincing and should be addressed. in this context, please note the paper from Gilles Charvin (Fehrmann et al., 2013). They used microfluidics to look at mitochondrial membrane potential in individual yeast cells as they aged. This paper should have been cited in the current manuscript, and it sets the bar both technically (for the microfluidic analysis of membrane potential and growth in individual cells) and conceptually (they found that decline of membrane potential is age-independent and leads to clonal senescence – i.e., is not reversible). The phenomenon studied in the current manuscript is different because it has to do with high-potential/low-respiration cells, but the Charvin study sets the precedent in the literature at least for irreversible mitochondrial-state changes.

We would like to thank the editor and the three reviewers for their constructive suggestions that helped improve our manuscript.

We have now added results of further analysis on ‘slow’ to ‘fast’ switching (Figure 5B-D and Figure 5—figure supplements 3-7). In addition, we also show transition of cells from high to low TMRE state and vice versa (Figure 5A and Figure 5—figure supplement 10). We have now cited the paper from the group of Gilles Charvin (Fehrmann et al., 2013) in the main text (subsection “Mitochondrial membrane potential but not amount predicts slow growth”, last paragraph).

We address the specific issues raised by each of the reviewers below.

Reviewer #1:Dhar et al. address an important topic of broad interest: cellular heterogeneity and its relationship to proliferation and survival. Given that very little is known about the causes of heterogeneity, their approach of doing a genetic screen for yeast mutants that alter the distribution of growth rates within a population is sensible, and the scale of the experiment (>1500 mutants) is impressive. However, despite my enthusiasm for the research question and the general approach, I do have some major concerns:1) The authors appear to have reinvented the micro-colony growth assay of Levy et al. (Levy et al., 2010) and should do a better job of presenting the similarities and differences between the two methods. Of particular concern is that they appear to have cell behaviors that were not seen by the other group – they mention lag (subsection “Image processing” and subsection “Cell sorting and growth measurement of sorted bins”, end of last paragraph) and nutrient deprivation, neither of which was seen by the other group (Levy et al., 2010 and Ziv et al., 2013) in medium with sufficient glucose.

We have now added more details to the subsection “Image processing” and show examples of lag and growth slowdown towards the end of the observation in Figure 5—figure supplement 4. We also note that lag was observed in the microcolony growth assay by Ziv et al. [Ziv et al., 2013].

2) The finding of a role for mitochondria in growth heterogeneity is interesting, but petites are well known to decrease growth rate and increase stress resistance. So the key advance here would be the finding that mitochondrial states switch back and forth, rather than petites just being a slow-growing dead end. Levy et al., 2010, had already demonstrated switching of growth states, and that single micro-colonies could give rise to the complete distribution of growth rates. So the question is whether the switching is driven by mitochondrial-state changes. Unfortunately, the experiments on switching are not entirely convincing. The authors took bins of cells sorted by TMRE (mitochondrial membrane potential) and assayed their growth. They found that high-TMRE cells tended to have slower growth rates and that high TMRE correlated with respiration deficiency. These results would be consistent with switching only in the direction of becoming petite. But the authors argue that switching can go in both directions because sorted bins could repopulate the full distributions of TMRE staining and growth. However, noise in TMRE staining or other technical reasons could mean that the sorted bins were contaminated to some extent with respiration-proficient cells, which then might be responsible for the repopulation of the distributions. A more direct tracking experiment is necessary.

We show that the contaminating cells in the HI bin cannot explain the percentage of low TMRE cells after 24 hours of growth (Figure 5A and Figure 5—figure supplement 10). We describe the calculations in the Materials and methods section (subsection “Switching of cells from high to low membrane potential”).

3) The manuscript does not offer much mechanistic insight into the changes in mitochondrial state or mtDNA copy number. What is causing these changes? Why do the changes persist for the number of cell generations that they do? Is the mitochondrial-state phenomenon tied to, or independent from, the heterogeneity in gene expression seen in Levy et al., 2010?

The underlying mechanism that drives changes in mitochondrial state and mtDNA copy number is indeed not clear. However, in Figure 4 we show that increased expression of the mitochondrial DNA polymerase Mip1 increased mtDNA copy number 3-fold (Figure 4B) and reduced both the fraction of slow proliferating and respiration-deficient cells and the fraction of cells with high TMRE signal (Figure 4C, D). This is consistent with defects in mtDNA replication causing the mitochondrial state and mtDNA copy number variation. The mitochondrial state phenomenon is likely to be different the from the heterogeneity in gene expression seen in Levy et al. 2010, as unlike Levy et al. 2010, we observed lower expression of TSL1 gene in our slow growing cells (Figure 6—figure supplement 2C).

Reviewer #2:[…] 1) The raw data of colony growth assays is not made available. This is not acceptable, as others may later want to explore the dataset in various ways. The raw data should be uploaded in a public repository in two forms:- Time-course data of the size of each colony, for all colonies and all yeast strains. This is important if others want to explore time-series for lag phase, saturations or other traits that may not be fully correlated to the growth rates computed by the authors.- The scalar values of growth-rates of individual colonies, so that others can use these values directly.

Data is available at https://github.com/lehner-lab/MicroscopyData_Dhar_et_al

2) GO analysis (Figure 2E) highlighted respiration based on many genes but little enrichment. Other terms were better enriched, although among fewer genes, especially for the 'red' set. The authors should mention in text what are the kinases, cytoskeletal binding proteins and carbohydrate transporters involved in maintaining the% slow fraction.

We have now mentioned some of these genes in the main text (subsection “Deletion of genes involved in mitochondrial function alter heterogeneity”, last paragraph).

Reviewer #3:[…] • S. cerevisiae is known to throw off petite cells that lack complete mtDNAs, in a genetic background dependent manner (these petites can also propagate in liquid cultures so multiple serial transfers can alter the initial number of petites in the inoculums which may influence the overall observed heterogeneity differences between strains). The "slow fraction" could represent these non-respiring petite cells, however the authors provide evidence that the "fast" to "slow" transitions are reversible (i.e. that the High TMRE, respiratory deficient cells could give rise to Low TMRE cells (subsection “Mitochondria state but not content predicts slow growth”, last paragraph, Figure 5—figure supplement 2). Have these "switched" cells regained respiratory capability? To decide if cells have switched between slow and fast growth, the authors rely on a threshold value of growth rates (as opposed to a percent change). It's not clear why the 0.3 h-1 threshold was chosen. As I understood, a microcolony that changes from 0.1 to 0.29 h-1 would be scored as slow growing while a microcolony that changes from 0.29 to 0.31 h-1 would be scored as one that switched between slow and fast growth. Can the authors clarify?

We have now addressed this issue in detail and we show that the results are robust to the cut off values that we use (Figure 5A-C and Figure 5—figure supplements 2 and 4). In addition, we have also added an analysis on bin switching where we divide the range of entire growth rate into 6 bins and calculate the percentage of cells switching from one bin to another (Figure 5—figure supplement 5). We also show that some cells lose respiration capability transiently and are able to regain this capability (Figure 5E).

• Yeast cells growing in microplates are likely to be oxygen starved (even if being shaken) and near transition states in metabolism, so it makes sense that changes in mitochondrial "states" are being observed. A discussion of how environmental conditions could influence heterogeneity would help to put this into context.

We have mentioned this now in the Discussion section (second paragraph).

[Editors' note: further revisions were requested prior to acceptance, as described below.]

The reviewers remained supportive of your study and appreciated most of the revisions. However, the evidences supporting the claim that cells switch from a low-growth to a fast-growing mode remained inconclusive. Since this is not the essence of your results, we are happy to accept the paper. however, we ask that you will back off this claim and move Figure 5 to the supplementary material.The reviewers also felt that the discussion of previous findings, and their relation to your work, will improve the manuscript. please try to do that as well. see below specific comments.

We thank the editor and the reviewers again for their constructive suggestions.

We have moved Figure 5 to the supplementary section (Figure 5—figure supplement 6, 7) and have minimized the discussion on switching in the manuscript.

We have also added a part discussing the relevance of our work in the context of already published work in the Discussion.

Reviewer #1:In this revised manuscript, Dhar et al. present new analyses aiming to address concerns about their inferences of switching from a slow-growing, respiration-deficient cell state to a fast-growing, respiration-capable cell state. I remain enthusiastic about the topic of the manuscript and I remain impressed by the scale of the mutant analysis, but I also remain concerned that the authors' analyses of switching rely on indirect arguments that are not entirely convincing. Specifically:1) They argue (subsection “Switching of cells from high to low membrane potential”) that TMRE-high to TMRE-low switching must take place to explain the observation that 1% of cells are TMRE-low immediately after sorting into the TMRE-high bin (a low rate of contamination), whereas 34.7% of cells are TMRE-low 24 hours later. Their argument relies on a simple model in which TMRE-high cells consist of two subpopulations (80% slow-growing and 20% fast-growing), TMRE-low cells are fast-growing, and the average slow and fast growth rates represent the corresponding subpopulations. This model has some problems. First, the arithmetic mean growth rate might not be appropriate to model changes in subpopulation sizes over time. If each cell chooses a growth rate at random from its distribution then what matters is the geometric mean, which is heavily influenced by low values. Alternatively, if growth rate is relatively stably inherited through time then what matters is the fastest members of a subpopulation. Either way, arithmetic mean could be a very poor indicator of subpopulation growth. Second, even if we ignore the problem with the arithmetic mean and take the authors' argument and apply it to the transition between the 24-hr time point and the 48-hr time point, then their calculation actually predicts quite well the 48-hr subpopulation sizes from the 24-hr starting point, so TMRE-high to TMRE-low switching is not required to explain this transition. Third, the authors' simple model ignores lag, which would likely further under-represent the slow-growing (high-TMRE) fraction, and might actually be why the transition from 0 to 24 hours requires additional explanation whereas the transition from 24 to 48 hours does not.

We agree with the reviewer that our model is quite simple and we could make it more complex by considering many more sub-populations with different growth rates. However, this model gives us a conservative estimate for switching of cells from high to low TMRE state.

The geometric mean growth rates of slow and fast sub-populations are 0.20 and 0.37 h^-1^ (as compared to arithmetic mean growth rate of 0.21 and 0.38 h^-1^ respectively). This does not substantially change our calculations of expected% of low TMRE cells without any switching from high to low TMRE state (subsection “Switching of cells from high to low membrane potential”). We did not consider any lag phase as it is difficult to determine the duration of the lag phase from our results and it seems that the lag phase is more pronounced in fast growing cells (Author response image 1). Nevertheless, including lag phase in our calculation would reduce the time for exponential growth for slow and fast-growing cells, which means there would be less time for fast growing low TMRE cells to take over the population. For example, in our calculation in the aforementioned subsection, if we consider 2 hours of lag phase for both slow and fast-growing cells, the expected% of low TMRE cells comes out to be 4.11% which is lower than what we expect without considering lag phase.

We also show calculations for expected% of low TMRE cells after 48 hours and again, the expected% of low TMRE cells is substantially lower than the observed% of low TMRE cells.

2) The authors have verified that their analyses of growth rate switching in micro-colonies are robust to changes in cutoffs between "fast" and "slow" growth, which is good, but what these switches mean is insufficiently discussed. In particular, fast-to-slow switches such as those shown in Figure 5D necessarily involve concerted changes in most cells in a micro-colony because a single cell changing from fast to slow in the middle of micro-colony growth would not appreciably change the growth rate of the micro-colony. The authors should discuss (and perhaps experimentally investigate) what such switches at the level of the micro-colony mean.

Indeed, we do not have the mechanistic understanding of the switching at the microcolony level and how many cells in a microcolony are switching their growth rates is not clear from our data. Hence, we have moved these results to the supplement (Figure 5—figure supplement 7, 8, 9) and have minimized discussion of switching in the manuscript.

3) The conclusion that some cells regain ability to respire is an interesting piece of this work but it is not clear exactly what the authors mean by it. In Figure 5E they present the "percentage of colonies that regained capability to respire" but the calculation involves two different plates that were not replicas, so the ability to "regain" respiration was a more indirect inference than the text suggests.Because of the above concerns, my view remains that more direct evidence of switching at the level of single cells (from high-TMRE to low-TMRE, from slow growth to fast growth, and from respiration-deficient to respiration capable) would substantially strengthen this paper.

The replica plating experiment for assessing the ability to regain respiration required us to know the time of switching which we were not able to determine through this experiment. Thus, we have removed this claim from the results part and have moved Figure 5 to the supplement (Figure 5—figure supplement 6, 7).

Reviewer #3:I completely agree that the resubmission only indirectly addresses our concerns about slow to fast growth. Even with the new statistical analysis, some quick growing contaminants could explain what they conclude are transitions. A simple replica plating experiment would have shown that their non-respiring cells can convert back to respiring cells. Given that they were previously asked for alternate experimentation to demonstrate the slow to fast transition, they need to back off this claim. (and minimize and move Figure 5 to supplemental).

We have moved Figure 5 to the supplementary material (Figure 5—figure supplement 6, 7) and also have minimized discussion on this topic in the manuscript.

In general, the authors could do better at discussing their findings in the context of previously published work.

We have now more extensively discussed our findings in the context of earlier work in the Discussion.

Their proposal (in the Discussion) that petites are part of a mitochondrial heterogeneity continuum is not substantiated if they are unable to convincingly show reversibility.

We have removed this proposal from the main text.

In the Abstract, they should change the word "developed" a high-throughput method to "used".

Done.

The authors now refer to mitochondrial membrane potential in places though there are still several places where mitochondrial "state" is used and could/should be corrected (e.g. Subsection “Mitochondrial membrane potential but not amount predicts slow growth”, fourth and fifth paragraphs, subsection “Variation in mitochondrial state predicts additional phenotypic heterogeneity including drug resistance”, first paragraph and especially in the first paragraph of the Discussion).

Changed now.